# Generalised Flow Maps for Few-Step Generative Modelling on Riemannian Manifolds

**Oscar Davis**[1*], **Michael S. Albergo**[2,3,4], **Nicholas M. Boffi**[5], **Michael M. Bronstein**[1,6],
**Avishek Joey Bose**[1,7,8]
[1]University of Oxford, [2]Harvard University, [3]Kempner Institute,
[4]Institute for Artificial Intelligence and Fundamental Interactions, MIT,
[5]Carnegie Mellon University, [6]AITHYRA, [7]Mila, [8]Imperial College London

## Abstract

Geometric data and purpose-built generative models on them have become ubiquitous in high-impact deep learning application domains, ranging from protein backbone generation and computational chemistry to geospatial data. Current geometric generative models remain computationally expensive at inference—requiring many steps of complex numerical simulation—as they are derived from dynamical measure transport frameworks such as diffusion and flow-matching on Riemannian manifolds. In this paper, we propose GENERALISED FLOW MAPS (GFM), a new class of few-step generative models that generalises the Flow Map framework in Euclidean spaces to arbitrary Riemannian manifolds. We instantiate GFMs with three self-distillation-based training methods: Generalised Lagrangian Flow Maps, Generalised Eulerian Flow Maps, and Generalised Progressive Flow Maps. We theoretically show that GFMs, under specific design decisions, unify and elevate existing Euclidean few-step generative models, such as consistency models, shortcut models, and meanflows, to the Riemannian setting. We benchmark GFMs against other geometric generative models on a suite of geometric datasets, including geospatial data, RNA torsion angles, and hyperbolic manifolds, and achieve state-of-the-art sample quality for single- and few-step evaluations, and superior or competitive log-likelihoods using the implicit probability flow.

## 1 Introduction

Dynamical measure transport offers a unifying and prescriptive framework for constructing neural network-based generative models that learn to sample a desired target distribution by pushing forward a tractable prior. Numerical solution of the resulting dynamical systems has led to popular method families—including diffusion models (Song & Ermon, 2020), flow matching (Lipman et al., 2022; Peluchetti, 2023; Liu et al., 2022; Albergo & Vanden-Eijnden, 2022), and stochastic interpolants (Albergo et al., 2023)—which together have revolutionised the field and led to state-of-the-art results over continuous modalities (Karras et al., 2024; Polyak et al., 2024). While often applied to Euclidean data such as images, this powerful paradigm naturally extends to data types that are inherently *geometric* and lie on a known Riemannian manifold, in which case the associated flows and diffusions are defined directly on the manifold (Huang et al., 2022b; De Bortoli et al., 2022; Chen & Lipman, 2024). Such geometric generative models have found widespread application in high-impact scientific settings such as rational drug design on the $SE(3)^N$ manifold of protein backbones (Bose et al., 2024; Huguet et al., 2024; Watson et al., 2023), generative material design in computational chemistry (Miller et al., 2024), and even discrete data using the Fisher-Rao geometry on the probability simplex (Davis et al., 2024; Cheng et al., 2025a).

The scalability of dynamical transport-based generative models arises from their use of simple regression-based objectives that lead to rapid *simulation-free* training. However, unlike training, obtaining high-quality generated samples at inference time requires numerical solution of the parametrised dynamical system, which necessitates evaluating the large learnt model numerous times. The computational complexity of inference is further burdened in geometric settings, where

---

*Correspondence to `oscar.davis@cs.ox.ac.uk`

each step of simulation requires computing potentially numerically unstable operators that must guarantee manifold constraints for faithful inference. For instance, for matrix Lie groups like $\mathrm{SO}(3)$, computing the standard exponential and logarithmic maps requires truncating an infinite matrix power series (Al-Mohy & Higham, 2012), which may incur additional sources of error beyond just step-size discretisation error compared to inference on Euclidean spaces. Therefore, reducing the number of Riemannian operations, while maintaining this powerful inductive bias, has the potential to lead to higher fidelity generated samples.

The search for accelerated inference with dynamical generative models defined on Euclidean spaces has spurred the development of fast inference techniques that fall broadly into two methodological families. The first family involves the intricate design of higher-order samplers (Dockhorn et al., 2022a; Zhang & Chen, 2022; Sabour et al., 2024; Karras et al., 2022), which can produce high-quality samples with less than ten steps. The second pursues direct learning of the flow map—the global solution of the governing differential equation—rather than the instantaneous vector field used in standard simulation (Boffi et al., 2024). This flow map perspective underlies distillation-based approaches, the most popular

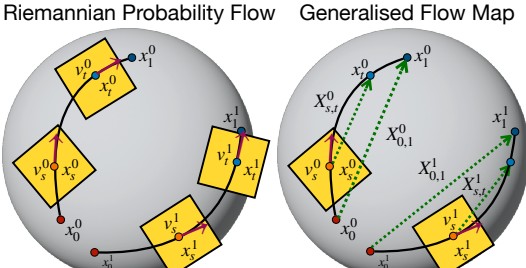

Figure 1: **Left:** The instantaneous vector field $v_t$ of the Riemannian probability flow ODE. **Right:** The Generalised flow map $X_{s,t}$ that jumps along the trajectory of the Riemannian probability flow ODE by steps of size $t - s$. Both models are depicted on $\mathbb{S}^2$.

of which include consistency models (Song et al., 2023b; Song & Dhariwal, 2023; Lu & Song, 2024), which attempt to learn the one-step flow map of a pre-trained teacher. Distillation-based methods have also been extended to the multi-step flow map setting we consider here at scale (Sabour et al., 2025). Recent advances have further demonstrated that state-of-the-art one-step generative models can be trained from scratch without a pre-trained teacher through self-consistent *self-distillation* (Boffi et al., 2025; Frans et al., 2025) or Monte Carlo direct training (Geng et al., 2025) objectives. These exciting developments naturally raise the question:

*Can the same flow map principles be extended to Riemannian settings, leading to a new class of few-step geometric generative models?*

Here, we answer this question in the affirmative by introducing GENERALISED FLOW MAPS (GFM), a method that generalises and extends flow maps (Boffi et al., 2025) to arbitrary Riemannian manifolds. Overall, our main contributions can be summarised as:

1. **Models**. We derive the technical conditions required for a flow map to respect manifold constraints, showing how they lead to three variants of GFM: Eulerian, Lagrangian, and Progressive.
2. **Learning**. We elevate the self-distillation losses presented in Boffi et al. (2025) to the Riemannian setting, showing how each variant can be learned efficiently, and how each recovers their Euclidean counterparts as a specific case.
3. **Empirical validation**. We comprehensively demonstrate the performance benefits of GFM over previous Riemannian generative models on standard geometric datasets. Specifically, we obtain state-of-the-art performance for single- and few-step sample quality on torsion angles found in RNA backbones and protein side-chains, geospatial data of natural disasters on the Earth's surface, synthetic data on $\mathrm{SO}(3)$ and on the Poincaré disk model of hyperbolic geometry, demonstrating our method's robustness to manifolds with non-trivial structures.

Our methodology opens the door to an under-explored design space of highly expressive generative models defined over geometric data that enjoy rapid inference.

## 2 BACKGROUND

### 2.1 RIEMANNIAN GEOMETRY

Informally, a $d$-dimensional manifold $\mathcal{M}$ is a topological space covered by a family of sets and corresponding maps, called *charts*, that render the manifold locally homeomorphic to $\mathbb{R}^d$. Requiring a $C^\infty$ differential structure on the charts makes the manifold *smooth*, and thereby naturally defines

the tangency of a vector $v \in \mathcal{T}_x\mathcal{M}$ to a point $x \in \mathcal{M}$, where $\mathcal{T}_x\mathcal{M}$ denotes the tangent space at point $x$, and the disjoint union of all tangent spaces $\mathcal{T}\mathcal{M} \coloneqq \bigsqcup_{x \in \mathcal{M}} \mathcal{T}_x\mathcal{M}$ forms the *tangent bundle*.

A *Riemannian manifold* is a pair $(\mathcal{M}, g)$, where $\mathcal{M}$ is a smooth manifold, and $g$ is a *metric tensor*, which smoothly assigns a bilinear and symmetric positive semi-definite inner product $g_x : \mathcal{T}_x\mathcal{M} \times \mathcal{T}_x\mathcal{M} \to \mathbb{R}$. This structure induces notions of curve lengths, distances, volumes, and angles. We assume $\mathcal{M}$ is connected, orientable, and complete, with volume element $\mathrm{d}x$. A *vector field* is a function from $\mathcal{M}$ to $\mathcal{T}\mathcal{M}$, such that $x \mapsto v(x) \in \mathcal{T}_x\mathcal{M}$. For a continuously differentiable map, $f : \mathcal{M} \to \mathcal{M}$, we define its *differential* at $x \in \mathcal{M}$, $\mathrm{d}f_x : \mathcal{T}_x\mathcal{M} \to \mathcal{T}_{f(x)}\mathcal{M}$, as the map given by $\mathrm{d}f_x : v \mapsto (f \circ \gamma)'(0)$ for *any* continuously differentiable curve $\gamma : [0, 1] \to \mathcal{M}$, such that $\gamma(0) = x$ and $\gamma'(0) = v$. The resulting map between tangent spaces is independent of $\gamma$.

**Manifold operations**. In the context of this work, we require a few usual operations on Riemannian manifolds. Specifically, we require a tangential projection from the ambient space, $\mathbb{R}^d$, to any arbitrary tangent plane $\mathcal{T}_x\mathcal{M}$ for $x \in \mathcal{M}$. The tangential projection enjoys a closed form: $P_x = I - \frac{n_x n_x^\top}{\|n_x\|^2}$, where $n_x \in \mathcal{T}_x\mathcal{M}$ is a basis vector. Roughly, the exponential map, $\exp_x : \mathcal{T}_x\mathcal{M} \to \mathcal{M}$, generalises the Euclidean notion of "$x + v$", and amounts to taking a step in the direction of the vector $v$. The logarithmic map, $\log_x : \mathcal{M} \to \mathcal{T}_x\mathcal{M}$, generalises "$q - x$" and provides a vector pointing towards its argument. Finally, the shortest path between two points $p, q \in \mathcal{M}$ is called a *geodesic* and is the curve $\gamma : [0, 1] \to \mathcal{M}$, with $\gamma(0) = p$ and $\gamma(1) = q$.

## 2.2 STOCHASTIC INTERPOLANTS AND FLOW MAPS

We are interested in the dynamical measure transport problem, which seeks to transport an easy-to-sample reference measure $\rho_0$ to a specified target measure $\rho_1$. As Riemannian manifolds can be embedded within a larger ambient space, $\mathcal{M} \subseteq \mathbb{R}^d$, we consider both $\rho_0, \rho_1 \in \mathcal{P}(\mathcal{M}) \subset \mathcal{P}(\mathbb{R}^d)$.

**Stochastic interpolants in $\mathbb{R}^d$.** In the context of generative modelling, a target distribution of interest is provided in the form of a dataset of samples, $\mathcal{D} = \{x_1^i\}_{i=1}^n$, where $x_1^i \stackrel{\text{iid}}{\sim} \rho_1(x_1)$, thus defining an empirical distribution, $\rho_{\text{data}}(x_1) \coloneqq \frac{1}{n} \sum_i^n \delta(x_1 - x_1^i)$. The goal in generative modelling is to build a model $\rho_1^\theta$, with parameters $\theta \in \Theta \subset \mathbb{R}^m$, that approximates $\rho_1^\theta \approx \rho_1$ in the distributional sense (*e.g.*, $W_2(\rho_1, \rho_1^\theta)$, or $\mathbb{D}_{\text{KL}}(\rho_1 \| \rho_1^\theta)$) while allowing us to draw new samples.

A modern and scalable approach for solving the dynamical measure transport problem—and by extension the generative modelling problem—is to leverage the framework of stochastic interpolants (Albergo et al., 2023). A stochastic interpolant is a stochastic process $I : [0, 1] \times \mathbb{R}^d \times \mathbb{R}^d \to \mathbb{R}^d$ that combines, in time, samples from the reference and target measures $(t, x_0, x_1) \mapsto I_t(x_0, x_1) = \alpha_t x_0 + \beta_t x_1$, for some choice of continuously differentiable functions $\alpha$ and $\beta$, such that $\alpha_0 = 1$, $\alpha_1 = 0$, $\beta_0 = 0$ and $\beta_1 = 1$. In addition, drawing pairs $(x_0, x_1) \sim \rho(x_0, x_1)$ from a given coupling must further satisfy the following marginal constraints $\int \rho(x_0, x_1)\mathrm{d}x_0 = \rho(x_1)$ and $\int \rho(x_0, x_1)\mathrm{d}x_1 = \rho(x_0)$. Connecting the measures $\rho_0$ and $\rho_1$, the interpolant defines a probability path, $(\rho_t)_{t \in [0,1]}$, which also follows the corresponding probability flow and transports particles using the following ordinary differential equation (ODE):

$$\partial_t x_t = \mathbb{E}_{x_t \sim \rho_t(x_t)}[\partial_t I_t \mid I_t = x_t] = v_t(x_t), \quad \text{with } x_0 \sim \rho_0(x_0). \tag{1}$$

Setting $\alpha_t \equiv 1 - t$ and $\beta_t \equiv t$ recovers well known methods such as flow-matching (Albergo & Vanden-Eijnden, 2022; Lipman et al., 2022) and rectified flow (Liu et al., 2022), while imposing optimal transport costs on the coupling results in OT-CFM (Tong et al., 2023).

**Flow maps in $\mathbb{R}^d$.** An alternative path to generating samples is to instead learn the integrator of the probability flow ODE directly, in order to avoid, at inference time, the costly numerical integration. Introduced in Boffi et al. (2024), flow maps are functions $X : [0, 1]^2 \times \mathbb{R}^d \to \mathbb{R}^d$ that satisfy the *jump condition*: $X_{s,t}(x_s) = x_t$, where $(x_t)_{t \in [0,1]}$ is any solution of the probability flow. One can thus sample from $\rho_1$ by first sampling $x_0 \sim \rho_0$ and then applying $X_{0,1}(x_0) \sim \rho_1$. These are parametrised as $X_{s,t}^\theta(x_s) = x + (t - s)v_{s,t}^\theta(x_s)$, for any $x$, $s$ and $t$, so that the boundary condition, at $t = s$, $X_{t,t}^\theta = \text{Id}$ is automatically satisfied. There are various self-distillation (*i.e.*, from scratch) objectives for learning a flow-map, which are used in conjunction with typical flow-matching objectives (Albergo et al., 2023; Tong et al., 2023; Lipman et al., 2023), For brevity, the three (Euclidean) flow map self-distillation loss variations—Eulerian, Lagrangian, and progressive—are provided in §A.

## 3 METHOD

We seek to define a generative model over arbitrary Riemannian manifolds that enables accelerated inference through few-step sampling. Towards this goal, we generalise the notion of flow maps to Riemannian manifolds, yielding GENERALISED FLOW MAPS (GFM) in §3.1. We then show how to train such GFM from scratch using self-distillation losses in §3.2, defined for arbitrary Riemannian manifolds—recovering the Euclidean case of Boffi et al. (2025) as a special case.

### 3.1 GENERALISED FLOW MAPS

We begin by defining a flow map in the context of Riemannian manifolds $\mathcal{M}$. To do so, we first recall that the flow map allows us to jump along the trajectory of the probability flow ODE connecting two measures $\rho_0, \rho_1 \in \mathcal{P}(\mathcal{M})$. The transport described by this ODE can be written in terms of the corresponding Riemannian flow and continuity equations:

$$\begin{aligned} \partial_t x_t &= v_t(x_t) \\ \partial_t \rho_t(x) &= -\mathrm{div}_g \left( \rho_t(x) v_t(x) \right), \end{aligned} \tag{2}$$

with $x_0 \sim \rho_0$, and where $\mathrm{div}_g$ denotes the Riemannian divergence induced by the metric $g$. As the probability flow ODE lives on $\mathcal{M}$, this presents an immediate point of departure from Euclidean spaces: the interpolant must trace a curve on $\mathcal{M}$. Therefore, the generalised interpolant, $I : [0,1] \times \mathcal{M}^2 \rightarrow \mathcal{M}$, satisfying the same endpoint constraints, $I_0(x_0, x_1) = x_0$ and $I_1(x_0, x_1) = x_1$, may not any more be any arbitrary linear combination of $x_0$ and $x_1$, as it may not belong to the manifold of interest. Consequently, we parametrise the interpolant as the geodesic connecting the points $x_0$ and $x_1$, $I : (t, x_0, x_1) \mapsto \exp_{x_0}(\alpha_t \log_{x_0}(x_1))$, with $\alpha_0 = 0$ and $\alpha_1 = 1$ (Chen & Lipman, 2024). This chosen form of $I_t$ also enables us to write the vector field of the flow in equation 2 as $\partial_t I_t(x_0, x_1) = \alpha'_t \log_{x_t}(x_1)/(1 - \alpha_t) \in \mathcal{T}_{x_t}\mathcal{M}$. Note that when the manifold is an Euclidean space (*i.e.*, $\mathcal{M} = \mathbb{R}^d$), we recover Albergo & Vanden-Eijnden (2022).

Given the stochastic interpolant, we now define GFM that jumps along the trajectory of equation 2.

**Definition 1.** *(GENERALISED FLOW MAPS) Let $(\mathcal{M}, g)$ be a Riemannian manifold, and let $\rho_0$ and $\rho_1$ be two distributions on $(\mathcal{M}, g)$. The generalised flow map is the unique function $X : [0,1]^2 \times \mathcal{M} \rightarrow \mathcal{M}$, such that, for any solution $(x_t)_{t \in [0,1]}$ of Equation (2), and any $(s, t) \in [0,1]^2$, $X_{s,t}(x_s) = x_t$.*

Analogous to the Euclidean case, GFM enables one-step sampling by first sampling $x_0 \sim \rho_0(x_0)$ and then applying $X_{0,1}(x_0) \sim \rho_1$. A natural parametrisation for constructing the GFM is $X_{s,t}(x_s) = \exp_{x_s}((t - s)v_{s,t}(x_s))$, with the underlying vector field $v : [0,1]^2 \times \mathcal{M} \rightarrow \mathcal{T}_{x_s}\mathcal{M}$. This automatically satisfies the boundary condition $X_{s,s}(x_s) = x_s$, since $\exp_{x_s}(\vec{0}) = x_s$. We can thus characterise a GFM in three different ways, generalising the Euclidean case.

**Proposition 1.** *Let $X_{s,t}$ be parametrised as $X_{s,t}(x) = \exp_x((t - s)v_{s,t}(x))$. Then $X_{s,t}$ is the unique GFM for equation 2 if and only if it satisfies any of the following conditions:*

*(i) Generalised Lagrangian Condition:*

$$\forall (s, t) \in [0,1]^2, x_s \in \mathcal{M}, \qquad \partial_t X_{s,t}(x_s) = v_t(X_{s,t}(x_s)), \tag{3}$$

*(ii) Generalised Eulerian Condition:*

$$\forall (s, t) \in [0,1]^2, x_s \in \mathcal{M} \qquad \partial_s X_{s,t}(x_s) + \mathrm{d}(X_{s,t})_{x_s}[v_s(x_s)] = 0, \tag{4}$$

*(iii) Generalised Semigroup Condition:*

$$\forall (s, t, u) \in [0,1]^3, x_s \in \mathcal{M}, \qquad X_{u,t}\left(X_{s,u}(x_s)\right) = X_{s,t}(x_s). \tag{5}$$

We include the proofs of this proposition in §B.1 alongside proofs of the legality of the claims. As stated in Proposition 1, the extension to Riemannian manifolds requires the use of manifold operations such as the differential in place of the Euclidean gradient. Moreover, instantaneously, the GFM recovers the vector field in its derivative and defines an implicit flow.

**Lemma 1.** *(Generalised Tangent Condition) Let $v_t(x_t) = \mathbb{E}_{x_t \sim \rho_t(x_t)}[\partial_t I_t \mid I_t = x_t]$ for any $t$ and $x_t \in \mathcal{M}$ be the drift of equation 2. Then it holds that $\lim_{s \to t} \partial_t X_{s,t}(x_s) = v_t(x_t)$.*

The proof for Lemma 1 is provided in §B.2 and also illustrated in Figure 1. The lemma underscores the key idea that, for $s = t$ (on the diagonal of the times space, $[0,1]^2$), the derivative of the GFM is the instantaneous vector field, $v_t$ of equation 2. Moreover, from the parametrisation in Proposition 1, and from the fact that $\lim_{s \to t} \partial_t X_{s,t}(x) = v_{t,t}(x)$ for any $x$, it follows that $v_{t,t} = v_t$ for any $t \in [0,1]$. (We thoroughly prove the limit in Lemma 4.) Therefore, we hereinafter use $v_{t,t}$ and $v_t$ interchangeably, and we may train the GFM on the diagonal $s = t$ using Riemannian Flow Matching (RFM) (Chen & Lipman, 2024), the loss of which is given by:

$$\mathcal{L}_{\mathrm{RFM}}(\theta) = \mathbb{E}_{(x_0,x_1) \sim \rho(x_0,x_1), x_t \sim \rho_t(x_t)} \left[ \| v_{t,t}^\theta(x_t) - \partial_t I_t(x_0,x_1) \|_g^2 \right]. \tag{6}$$

### 3.2 GENERALISED SELF-DISTILLATION LOSSES

Given the above characterisations of the GFM in Proposition 1, we define three new generalised *self-distillation* objectives, leading to Generalised Lagrangian self-distillation (G-LSD), Generalised Eulerian self-distillation (G-ESD), and Generalised progressive self-distillation (G-PSD). Specifically, we consider self-distillation objectives of the following form:

$$\mathcal{L}(\theta) = \mathcal{L}_{\mathrm{RFM}}(\theta) + \mathcal{L}_{\mathrm{GFM\text{-}SD}}(\theta). \tag{7}$$

To adapt to the Riemannian setting, we further constrain our vector field $v^\theta$ to lie on the tangent plane at the point where it is evaluated. Specifically, letting $f^\theta : [0,1]^2 \times \mathcal{M} \to \mathbb{R}^d$ be our underlying neural network, then, for any $0 \leq s \leq t \leq 1$, $p \in \mathcal{M}$, $v_{s,t}^\theta(p) := \mathrm{proj}_{\mathcal{T}_p \mathcal{M}} \left( f_{s,t}^\theta(p) \right)$. This ensures that all the usual Riemannian manifold operations required are not ill-defined.

**Generalised Lagrangian self-distillation**. As linear combination of vectors also belong to the same tangent space, we may freely consider the difference between $X_{s,t}^\theta(I_s)$ and $v_{t,t}^\theta(X_{s,t}^\theta(I_s))$. However, we further adapt the loss to incorporate the Riemannian metric in the norm of the resulting vector so that the losses are comparable with the flow matching ones (namely in terms of magnitude).

**Proposition 2.** *(Generalised Lagrangian self-distillation) The* GFM *is the unique minimiser of the objective in equation 7 for $\mathcal{L}_{\mathrm{GFM\text{-}SD}}(\theta) = \mathcal{L}_{\mathrm{G\text{-}LSD}}(\theta)$, where*

$$\mathcal{L}_{\mathrm{G\text{-}LSD}}(\theta) = \mathbb{E}_{t,s,(x_0,x_1)} \left[ \left\| \partial_t X_{s,t}^\theta(I_s) - v_{t,t}^\theta(X_{s,t}^\theta(I_s)) \right\|_g^2 \right]. \tag{8}$$

The proof for Proposition 2 is provided in §B.3.

**Eulerian self-distillation**. In an analogous manner to the Generalised Lagrangian self-distillation loss, we can instantiate an Eulerian loss to the Riemannian setting.

**Proposition 3.** *(Generalised Eulerian self-distillation) The* GFM *is the unique minimiser of the objective in equation 7 for $\mathcal{L}_{\mathrm{GFM\text{-}SD}}(\theta) = \mathcal{L}_{\mathrm{G\text{-}ESD}}(\theta)$, where*

$$\mathcal{L}_{\mathrm{G\text{-}ESD}}(\theta) = \mathbb{E}_{t,s,(x_0,x_1)} \left[ \left\| \partial_s X_{s,t}^\theta(x_s) + \mathrm{d}(X_{s,t}^\theta)_{I_s}[v_{s,s}^\theta(I_s)] \right\|_g^2 \right]. \tag{9}$$

For completeness, the proof for Proposition 3 is included in §B.3.

**Progressive self-distillation**. Naturally, $X_{u,t} \circ X_{s,u} : \mathcal{M} \to \mathcal{M}$ is well-defined for all $(s,u,t) \in [0,1]^3$. Thus, we may use the geodesic distance, $d_g$, to derive the Generalised Progressive Self-Distillation (G-PSD) objective to enforce the semigroup condition equation 5. The G-PSD objective is the simplest to port, as it is devoid of any spatial or time derivatives of the GFM as stated in the following proposition. The proof is deferred to §B.3.

**Proposition 4.** *(Generalised Progressive Self-Distillation) The* GFM *is the unique minimiser over $v_\theta$ of equation 7 for $\mathcal{L}_{\mathrm{GFM\text{-}SD}}(\theta) = \mathcal{L}_{\mathrm{G\text{-}PSD}}(\theta)$ and $u \mid (s,t) \sim \mathcal{U}(s,t)$, where*

$$\mathcal{L}_{\mathrm{G\text{-}PSD}}(\theta) = \mathbb{E}_{t,s,u,(x_0,x_1)} \left[ d_g^2 \left( X_{s,t}^\theta(I_s), X_{u,t}^\theta \left( X_{s,u}^\theta(I_s) \right) \right) \right]. \tag{10}$$

---

**Algorithm 1:** GFM training, for any choice of self-distillation.

---

**Input:** Riemannian manifold, $(\mathcal{M}, g)$; time distributions, $T$, $S \mid T$; coupling, $\rho$; batch size, $M$.
**repeat**

Draw batch $(t_i, s_i, x_0^i, x_1^i)_{i=1}^M \sim (T, S \mid T, \rho(x_0, x_1))$;

Construct $x_s^i = I(x_0^i, x_1^i, s^i)$, compute $u_s^i = \partial_{s^i} I(x_0^i, x_1^i, s^i)$;

Estimate $\hat{\mathcal{L}}_{\text{RFM}}(\theta, (x_s^i)_{i=1}^M, (u_s^i)_{i=1}^M) \approx \mathcal{L}_{\text{RFM}}(\theta)$;

Estimate $\hat{\mathcal{L}}_{\text{GFM-SD}}(\theta, x_s^i, s^i, t^i) \approx \mathcal{L}_{\text{GFM-SD}}(\theta)$;

Optimisation step on $\hat{\mathcal{L}}_{\text{RFM}}(\theta, (x_s^i)_{i=1}^M, (u_s^i)_{i=1}^M) + \hat{\mathcal{L}}_{\text{GFM-SD}}(\theta, (x_s^i)_{i=1}^M, (s^i)_{i=1}^M, (t^i)_{i=1}^M)$;

**until** *converged*;
**Output:** Flow map $X^\theta$.

---

### 3.3 TRAINING GENERALISED FLOW MAPS

While it is possible to implement the GFM objectives naively, it may incur unstable training dynamics. Instead, and in line with the literature (Sabour et al., 2025; Boffi et al., 2025; Geng et al., 2025), we opt for a *self-bootstrapped* objective by placing a stop-gradient operator to only optimise parts of each objective. This, in turn, converts one term in the objective as the "teacher", which is distilled to terms in the loss that have non-zero parameter gradients. We include in Algorithm 1 the pseudocode used to train a GFM for any self-distillation loss.

**G-LSD loss**. We place the stop-gradient on the second term of equation 3, yielding the objective:

$$\hat{\mathcal{L}}_{\text{G-LSD}}(\theta) = \mathbb{E}_{t,s,(x_0,x_1)} \left[ \left\| \partial_t X_{s,t}^\theta(I_s) - \text{stopgrad}\left( v_{t,t}^\theta(X_{s,t}^\theta(I_s)) \right) \right\|_g^2 \right]. \tag{11}$$

Back-propagating through $\partial_t X_{s,t}^\theta$ is simple in usual modern ML libraries, as forward-mode automatic differentiation is typically available alongside the Jacobian-Vector Product (JVP) operation.

**G-ESD loss**. Similarly, for the Eulerian loss, we apply the stop-gradient to the second term, which contains the spatial derivative—and thus avoiding higher order derivatives—leading to:

$$\hat{\mathcal{L}}_{\text{G-ESD}}(\theta) = \mathbb{E}_{t,s,(x_0,x_1)} \left[ \left\| \partial_s X_{s,t}(I_s) + \text{stopgrad}\left( d(X_{s,t}^\theta)_{I_s}[v_{s,s}^\theta(I_s)] \right) \right\|_g^2 \right]. \tag{12}$$

Note that this results in an objective closely related to that of (a Riemannian generalisation of) Mean Flows (Geng et al., 2025)—a connection we prove in §B.4. We also implement the latter, and term this objective G-MF (Generalised Mean Flows), and define it fully in §B.4. This objective is trained as in Mean Flows; that is to say, without the flow matching loss.

**G-PSD loss**. Finally, we also utilise the stop-gradient operator on the PSD objective using the two, smaller steps ($s$ to $u$, $u$ to $t$) as the teacher for the larger step ($s$ to $t$), resulting in the following:

$$\hat{\mathcal{L}}_{\text{G-PSD}}(\theta) = \mathbb{E}_{t,s,u,(x_0,x_1)} \left[ d_g^2 \left( X_{s,t}^\theta(I_s), \text{stopgrad}\left( X_{u,t}^\theta \left( X_{s,u}^\theta(I_s) \right) \right) \right) \right]. \tag{13}$$

We follow Boffi et al. (2025) by setting $u = \frac{1}{2}s + \frac{1}{2}t$, leading to two half-steps, thus generalising shortcut models of Frans et al. (2025) to the manifolds—a connection we also detail in §B.4.

## 4 EXPERIMENTS

We test the empirical calibre of GFM on a suite of standard geometric generative modelling benchmarks. Specifically, we instantiate GFM on torsion angles ($\mathbb{T}^2 \cong \mathbb{S}^1 \times \mathbb{S}^1$) found in protein side chains and RNA backbones $\mathbb{T}^7 \cong (\mathbb{S}^1)^7$ (Lovell et al., 2003; Murray et al., 2003), catastrophic geospatial events on Earth ($\mathbb{S}^2$) as introduced in Mathieu & Nickel (2020a), a synthetic dataset of the manifold of 3D rotations (SO(3)), and on the Poincaré disk (hyperbolic geometry).

**Metrics**. To evaluate the few-step sample quality of our models, we use the empirical MMD (Maximum Mean Discrepancy) between the test-set and the samples, with an RBF kernel using the manifold's distance, $d_g$, and a bandwidth of $\kappa = 1$. See §C for the calculation details. To assess the learnt vector field, we also compute the negative log-likelihood (NLL) on the test set when

Table 1: Test NLL on protein sidechain and RNA torsion angles. Standard deviation estimated over 5 runs. [†] indicates baseline numbers taken from Huang et al. (2022b).

|  | General (2D) | Glycine (2D) | Proline (2D) | Pre-Pro (2D) | RNA (7D) |
|---|---|---|---|---|---|
| **Dataset size** | 138,208 | 13,283 | 7,634 | 6,910 | 9,478 |
| MoPS[†] | 1.15±0.002 | 2.08±0.009 | 0.27±0.008 | 1.34±0.019 | 4.08 ± 0.368 |
| RDM[†] (Huang et al., 2022b) | 1.04±0.012 | 1.97±0.012 | 0.12±0.011 | 1.24 ± 0.004 | −3.70±0.592 |
| RFM (Chen & Lipman, 2024) | 1.01 ± 0.025 | **1.90 ± 0.055** | 0.15 ± 0.027 | 1.18 ± 0.055 | **−5.20 ± 0.067** |
| G-LSD (ours) | 0.99 ± 0.05 | 1.99 ± 0.02 | 0.24 ± 0.07 | 1.11 ± 0.02 | −4.15 ± 0.09 |
| G-PSD (ours) | **0.95 ± 0.02** | 1.94 ± 0.03 | **0.08 ± 0.04** | 1.10 ± 0.04 | −4.40 ± 0.13 |
| G-ESD (ours) | 0.99 ± 0.04 | 1.95 ± 0.01 | 0.19 ± 0.04 | 1.10 ± 0.02 | −4.61 ± 0.07 |
| G-MF (ours) | 0.97 ± 0.01 | 1.97 ± 0.01 | 0.21 ± 0.04 | **1.02 ± 0.04** | −3.79 ± 0.09 |

Table 2: MMD for 1 NFE with the test-set for proteins torsion angles and RNA backbones. Standard deviation estimated over 5 seeds.

|  | General (2D) | Glycine (2D) | Proline (2D) | Pre-Pro (2D) | RNA (7D) |
|---|---|---|---|---|---|
| RFM (Chen & Lipman, 2024) | 0.45 ± 0.006 | 0.27 ± 0.008 | 0.52 ± 0.057 | 0.47 ± 0.022 | 0.68 ± 0.011 |
| G-LSD (ours) | **0.02 ± 0.003** | **0.03 ± 0.004** | **0.04 ± 0.012** | **0.05 ± 0.004** | **0.08 ± 0.007** |
| G-PSD (ours) | 0.11 ± 0.016 | 0.05 ± 0.019 | 0.07 ± 0.011 | 0.08 ± 0.015 | 0.14 ± 0.027 |
| G-ESD (ours) | 0.29 ± 0.002 | 0.13 ± 0.006 | 0.44 ± 0.024 | 0.26 ± 0.016 | 0.45 ± 0.006 |
| G-MF (ours) | 0.11 ± 0.029 | 0.04 ± 0.019 | 0.09 ± 0.046 | 0.09 ± 0.017 | 0.20 ± 0.019 |

available. We have included in Appendix C.1 a discussion explaining why NLL is relevant in this context. Finally, we provide qualitative samples to further assert the method's soundness.

**Baselines**. We use RFM (Chen & Lipman, 2024), which is the state-of-the-art method, as the main baseline. Additionally, we include results for a Riemannian diffusion model (RDM) (Huang et al., 2022b) and Riemannian score-based generative models (RSGM) (De Bortoli et al., 2022) with test NLL results taken directly from the respective papers due to a lack of open source code. We also report a mixture of power spherical distributions (MoPS) (De Cao & Aziz, 2020) for test NLL.

Finally, we also detailed the computational cost of having using our proposed methods against RFM in Appendix C.4.

### 4.1 PROTEINS TORSION ANGLES AND RNA BACKBONES ON FLAT TORI

We train our model on a protein dataset on the flat 2D and 7D (RNA) tori ($\mathbb{S}^1 \times \mathbb{S}^1$ and $(\mathbb{S}^1)^7$). The 2D data is from Lovell et al. (2003) and the 7D data from Murray et al. (2003), compiled together by Huang et al. (2022a). We report our MMD results in Tables 1 and 2 and qualitative results in Figures 2 and 3. We observe that the test NLLs produced by GFM outperform those of RFM for the protein side chain datasets, and are marginally worse on the RNA dataset. Most importantly, we test the *one-step* generative capability that is unique to GFM in Table 2 and Figure 3 and find that GFM offers considerable gains, especially G-LSD, which offers an improvement of up to $22\times$ on the MMD (on "General") for a single function evaluation. We also plot Ramachandran plots on the torsion angles in Figure 2, where we observe that our methods achieve log-likelihood landscapes comparable to those of RFM. Finally, as an ablation, we also plot in Figure 3 the MMD as a function of inference steps and find that GFM consistently improve on the MMD for low NFEs, with quality consistently improving with higher steps as RFM.

Figure 2: Ramachandran plots on the General protein dataset. Test-set samples depicted in red.

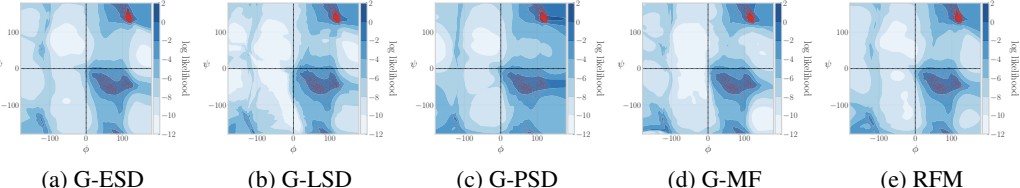

| (a) G-ESD | (b) G-LSD | (c) G-PSD | (d) G-MF | (e) RFM |

Figure 3: MMD on protein datasets against the NFE.

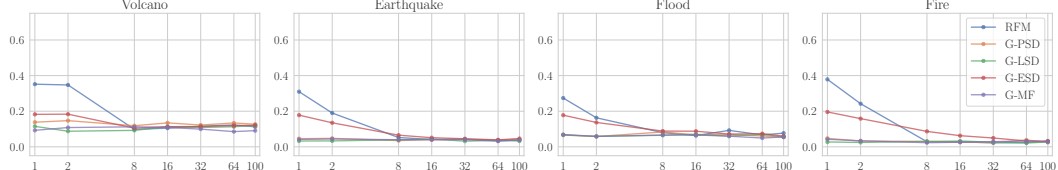

## 4.2 EARTH CATASTROPHES ON THE 2-SPHERE

We evaluate GFM on a collection of geospatial data, represented on the 2-sphere, $\mathbb{S}^2$. The dataset was first introduced in Mathieu & Nickel (2020b) and is curated from various sources (NOAA, 2020a;b; Brakenridge, 2017; EOSDIS, 2020). We report our results in Figure 4 and Table 3. Analogous to tori, we observe that the implicit flow $v_{t,t}^{\theta}$ within GFM offers log-likelihoods that outperform those of RFM and all other methods on three out of four datasets, with the exception of "Volcanoes". To assess sample quality, we again plot the MMD as a function of the number of integration steps, and observe once more great improvement on MMD for all our methods at 1 or 2 NFEs, while preserving high quality for higher NFEs. Finally, we provide the densities learnt by RFM and G-LSD on 2D Earth plots in Figure 5 (with the remaining plots in Figure 7), and remark similar likelihood landscapes between RFM and G-LSD.

Table 3: Test NLL on the Earth datasets. Standard deviation is estimated over 5 runs. † indicates baseline numbers taken from Huang et al. (2022b).

|  | **Volcano** | **Earthquake** | **Flood** | **Fire** |
|---|---|---|---|---|
| **Dataset size** | 827 | 6,120 | 4,875 | 12,809 |
| Mixture of Kent† | $-0.80\pm0.47$ | $0.33\pm0.05$ | $0.73\pm0.07$ | $-1.18\pm0.06$ |
| Riemannian CNF† (Mathieu & Nickel, 2020a) | $-0.97\pm0.15$ | $0.19\pm0.04$ | $0.90\pm0.03$ | $-0.66\pm0.05$ |
| Moser Flow† (Rozen et al., 2021) | $-2.02\pm0.42$ | $-0.09\pm0.02$ | $0.62\pm0.04$ | $-1.03\pm0.03$ |
| Stereographic Score-Based† | $-4.18\pm0.30$ | $-0.04\pm0.11$ | $1.31\pm0.16$ | $0.28\pm0.20$ |
| RSGM† (De Bortoli et al., 2022) | $-5.56\pm0.26$ | $-0.21\pm0.03$ | $0.52\pm0.02$ | $-1.24\pm0.07$ |
| RDM† (Huang et al., 2022b) | $-6.61\pm0.97$ | $-0.40\pm0.05$ | $0.43\pm0.07$ | $-1.38\pm0.05$ |
| RFM (Chen & Lipman, 2024) | $\mathbf{-7.93 \pm 1.67}$ | $-0.28 \pm 0.08$ | $0.42 \pm 0.05$ | $-1.86 \pm 0.11$ |
| G-LSD (ours) | $-4.96 \pm 0.68$ | $-0.93 \pm 0.01$ | $-0.38 \pm 0.33$ | $-2.14 \pm 0.42$ |
| G-PSD (ours) | $-3.50 \pm 0.22$ | $-0.63 \pm 0.13$ | $-0.76 \pm 0.13$ | $\mathbf{-2.48 \pm 0.71}$ |
| G-ESD (ours) | $-4.49 \pm 0.20$ | $-0.67 \pm 0.08$ | $\mathbf{-0.88 \pm 0.38}$ | $-2.29 \pm 0.08$ |
| G-MF (ours) | $-3.73 \pm 0.41$ | $\mathbf{-1.08 \pm 0.09}$ | $-0.72 \pm 0.11$ | $-2.24 \pm 0.30$ |

Figure 4: MMD on Earth datasets against the NFE.

## 4.3 SO(3) SYNTHETIC DATA

We next instantiate GFM on the manifold of 3D rotations, SO(3), using the synthetic dataset from Huang et al. (2022a). We compute both test NLL and MMD values and report them in Table 4. We find that, on test NLL, all methods perform roughly equally, and there is no clear winner. On MMD, we find that all versions of GFM outperform RFM, with G-LSD being again the most performant. This demonstrates the effectiveness of all our methods on non-trivial manifolds.

Figure 5: Plots of densities for the various datasets and all compared methods. Depicted in red are the test-set samples. Datasets from left to right: volcano, earthquake, flood, fire.

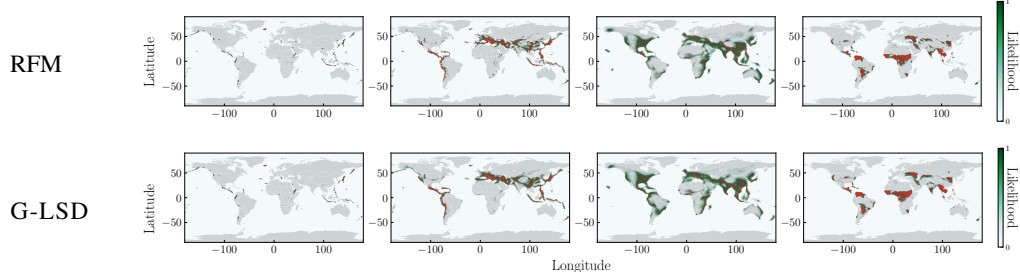

Figure 6: MMD on the hyperbolic dataset against the NFE.

Table 4: Results on the $SO(3)$ test-set with standard deviation estimated over 5 seeds.

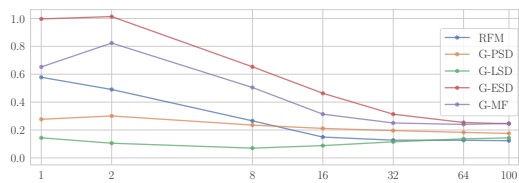

|  | MMD | | | NLL |
|---|---|---|---|---|
|  | 1 NFE | 2 NFE | 100 NFE |  |
| RFM | $0.147 \pm 0.007$ | $0.083 \pm 0.003$ | $0.042 \pm 0.002$ | $-7.15 \pm 0.03$ |
| G-LSD (ours) | $\mathbf{0.064 \pm 0.007}$ | $\mathbf{0.059 \pm 0.005}$ | $0.044 \pm 0.008$ | $-7.11 \pm 0.03$ |
| G-PSD (ours) | $0.121 \pm 0.01$ | $0.073 \pm 0.005$ | $\mathbf{0.039 \pm 0.004}$ | $-7.15 \pm 0.03$ |
| G-ESD (ours) | $0.411 \pm 0.001$ | $0.408 \pm 0.001$ | $0.109 \pm 0.011$ | $\mathbf{-7.20 \pm 0.02}$ |
| G-MF (ours) | $0.291 \pm 0.007$ | $0.280 \pm 0.015$ | $0.283 \pm 0.014$ | $-6.85 \pm 0.03$ |

## 4.4 HYPERBOLIC MANIFOLDS

We evaluate GFM on a manifold with a non-trivial metric, in particular, on the Poincaré ball for hyperbolic geometry. We draw 20,000 samples from a target distribution that is wrapped normal, which we then use to train all methods, including RFM. We report the MMD as a function of the NFEs in Figure 6, and observe that all versions of GFM outperform RFM, even at high NFE, except for G-MF and G-ESD. Indeed, it seems that the variance during training was higher than for the other methods, which caused it to under-fit the distribution at lower time steps.

## 5 RELATED WORK

**Riemannian generative models**. The most related early efforts to build manifold structure into generative models come from conventional normalising flows built out of iterative coupling layers (Tabak & Vanden-Eijnden, 2010; Rezende & Mohamed, 2016; Dinh et al., 2017), where each coupling layer was designed to preserve the manifold structure (Bose et al., 2020; Rezende et al., 2020; Kanwar et al., 2020; Boyda et al., 2021; Bose et al., 2021). These ideas have also been extended to continuous-time flows and diffusions on general Riemannian structures (Lou et al., 2020; Mathieu & Nickel, 2020a; Falorsi & Forré, 2020; Bortoli et al., 2022), and their optimisation has also been made simulation-free (Rozen et al., 2021; Chen & Lipman, 2024; Bose et al., 2020; Davis et al., 2024). Here, our focus is on extending these approaches so that the more efficient any-step flow map is well-posed and learnable on general geometries.

**Accelerated inference in generative models**. Early work on accelerated inference focused on a teacher-student procedure (Song et al., 2020a; Luhman & Luhman, 2021; Salimans & Ho, 2022; Meng et al., 2023) where an expensive inference model is distilled to produce the same output in fewer steps. Orthogonal to this have been efforts to parallel diffusion inference with adaptive (Chen et al., 2024; Dockhorn et al., 2022b; Shih et al., 2023; Tang et al., 2024) or speculative (Bortoli et al., 2025) schemes. Consistency models aim to directly learn the one-step map to the data distribution from any point along the trajectory (Song et al., 2023b; Kim et al., 2024; Song & Dhariwal, 2023). The flow map (Boffi et al., 2024) has emerged as a unifying picture, and recent efforts have shown how to distill it (Sabour et al., 2024) or directly learn it (Boffi et al., 2025; Geng et al., 2025) using the equations that characterise it. We take the leap here to show how to generalise this complete class of models to the Riemannian setting for performant ends.

**Concurrent work**. Concurrent and most related to our work is that of Cheng et al. (2025b). The authors propose a new method, Riemannian Consistency Models (RCM), to train few-step generative models on Riemannian manifolds from scratch. Their approach directly ports the original Consistency Models (Song et al., 2023b) to Riemannian geometries, relying on more sophisticated geometric constructions. In contrast, GFM enjoys simpler practical instantiations as it relies on the self-distillation of the Flow Maps framework (Boffi et al., 2025) and recovers shortcut models as a special case of the PSD objective executed on Riemannian manifolds.

## 6 CONCLUSION

We propose GENERALISED FLOW MAPS, a new class of geometric generative models that are capable of performing few-step inference on arbitrary Riemannian manifolds. To build GFM, we provide three equivalent theoretical conditions that characterise a flow map on manifolds, and three corresponding self-distillation objectives. We demonstrate the empirical performance of GFM in the low NFE regime and achieve state-of-the-art results in sample-based metrics, and competitive test likelihoods in comparison to RFM. While each GFM condition leads to a different corresponding objective, at present, the Lagrangian objective remains the most performant in generating high-quality samples, and understanding this empirical observation from a theoretical lens is a natural direction for future work. Additionally, as we empirically demonstrate, the implicit flow within the flow map may lead to better NLL than the flow learned through RFM, which points to an interesting direction for gaining theoretical understanding in future work.

## ACKNOWLEDGEMENTS

OD is funded by both Project CETI and Intel. AJB is partially supported by an NSERC Post-doc fellowship. This research is partially supported by the EPSRC Turing AI World-Leading Research Fellowship No. EP/X040062/1 and EPSRC AI Hub No. EP/Y028872/1. MSA is supported by a Junior Fellowship at the Harvard Society of Fellows as well as the National Science Foundation under Cooperative Agreement PHY-2019786 (The NSF AI Institute for Artificial Intelligence and Fundamental Interactions, http://iaifi.org/). This work has been made possible in part by a gift from the Chan Zuckerberg Initiative Foundation to establish the Kempner Institute for the Study of Natural and Artificial Intelligence.

## ETHICS STATEMENT

We hereby acknowledge and declare to have abided by the ICLR Code of Ethics. Our current work does not utilise any sensitive data, nor does it directly enable nefarious usage, although other derivative works could use ours in unethical contexts. The authors declare no competing financial interests.

## REPRODUCIBILITY STATEMENT

All of our results are reproducible and have been run with the same 5 random seeds that were set in advance, and that were never changed throughout our evaluation, to provide us our standard deviations. The datasets we use are entirely public and are freely available to all. Our code will be made public upon acceptance alongside detailed instructions for running it locally, and the exact configuration files that were used.

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

# A  SELF-DISTILLATION ON EUCLIDEAN SPACES

In this section, we provide a self-contained review of the three self-distillation based objective functions introduced by Boffi et al. (2025). These extend upon the original distillation-based objectives introduced in Boffi et al. (2024). To this end, we first recall three key properties of the flow map.

**Proposition 5** (Flow map characterization). *Let $\partial_t x_t = v_t(x_t)$ denote a probability flow ODE, and let $X_{s,t} : [0,1]^2 \to \mathbb{R}^d$ denote its flow map. Then,*

i. *$X_{s,t}$ is the unique solution to the Lagrangian equation,*

$$\partial_t X_{s,t}(x) = v_t(X_{s,t}(x)) \qquad \forall\, (x,s,t) \in \mathbb{R}^d \times [0,1]^2. \tag{14}$$

ii. *$X_{s,t}$ is the unique solution to the Eulerian equation,*

$$\partial_s X_{s,t}(x) + \nabla X_{s,t}(x) v_s(x) = 0 \qquad \forall\, (x,s,t) \in \mathbb{R}^d \times [0,1]^2. \tag{15}$$

iii. *$X_{s,t}$ satisfies the semigroup condition,*

$$X_{s,t}(x) = X_{u,t}(X_{s,u}(x)) \qquad \forall\, (x,s,u,t) \in \mathbb{R}^d \times [0,1]^3. \tag{16}$$

*Proof.* We simply prove the forward direction: that the flow map satisfies each property. The proof of uniqueness follows similarly and can be found in Boffi et al. (2025).

Each property follows from the defining condition

$$X_{s,t}(x_s) = x_t \qquad \forall\, (s,t) \in [0,1]^2. \tag{17}$$

Taking the derivative with respect to time, we find that

$$\begin{aligned} \partial_t X_{s,t}(x_t) &= \partial_t x_t, \\ &= v_t(x_t), \\ &= v_t(X_{s,t}(x_s)), \end{aligned} \tag{18}$$

which is the Lagrangian equation. Noting that $x_s$ was arbitrary completes the proof.

Taking the total derivative of equation 17 with respect to $s$, we find

$$\begin{aligned} \partial_s X_{s,t}(x_s) + \nabla X_{s,t}(x_s)\partial_t x_s &= 0, \\ \implies \partial_s X_{s,t}(x_s) + \nabla X_{s,t}(x_s) b_s(x_s) &= 0. \end{aligned} \tag{19}$$

Again, noting that $x_s$ was arbitrary completes the proof.

The semigroup condition follows simply by noting that

$$X_{u,t}(X_{s,u}(x_s)) = X_{u,t}(x_u) = x_t = X_{s,t}(x_s). \tag{20}$$

This completes the proof. $\qquad\square$

As discussed in the main text, from the Lagrangian condition in equation 14, we may observe that

$$\lim_{s\to t} \partial_t X_{s,t}(x) = \lim_{s\to t} v_t(X_{s,t}(x)) = v_t(x) \tag{21}$$

assuming continuity of $X$ and using that $X_{t,t}(x) = x$ for all $(x,t) \in \mathbb{R}^d \times [0,1]$. Moreover, using the parameterization

$$X_{s,t}(x) = x + (t-s)v_{s,t}(x) \tag{22}$$

we find that

$$\lim_{s\to t} \partial_t X_{s,t}(x) = \lim_{s\to t} \left\{ (t-s)\partial_t v_{s,t}(x) + v_{s,t}(x) \right\} = v_{t,t}(x) = v_t(x). \tag{23}$$

This observation immediately leads to three "self-distillation" schemes, each of which proceeds by training the flow $v_{t,t}(x)$ on the diagonal $s = t$ via flow matching, and simultaneously distilling it into a flow map by minimizing the square residual on one of the conditions in Proposition 5.

**Proposition 6** (Euclidean self-distillation (Boffi et al., 2025)). *Let $X_{s,t}^{\theta}(x) = x + (t-s)v_{s,t}^{\theta}(x)$ denote a candidate flow map, and assume that $v^{\theta}$ is continuous in both time arguments. Moreover, let*

$$\mathcal{L}_b(v^{\theta}) = \int_0^1 \mathbb{E}\left[\left|v_{t,t}^{\theta}(I_t) - \partial_t I_t\right|^2\right] dt \qquad (24)$$

*denote the flow matching loss on the diagonal $s = t$. Then, the ideal flow map $X_{s,t}(x) = x + (t-s)v_{s,t}(x)$ corresponds to the unique minimizer over $v^{\theta}$ of each of the following objective functions.*

    *i. The Lagrangian self-distillation loss,*

$$\mathcal{L}_{\mathsf{LSD}}(\hat{v}) = \mathcal{L}_b(v^{\theta}) + \int_0^1 \int_0^t \mathbb{E}\left[\left|\partial_t X_{s,t}^{\theta}(I_s) - v_{t,t}^{\theta}(X_{s,t}^{\theta}(I_s))\right|^2\right] ds dt, \qquad (25)$$

    *ii. The Eulerian self-distillation loss,*

$$\mathcal{L}_{\mathsf{ESD}}(\hat{v}) = \mathcal{L}_b(v^{\theta}) + \int_0^1 \int_0^t \mathbb{E}\left[\left|\partial_s X_{s,t}^{\theta}(I_s) + \nabla X_{s,t}^{\theta}(I_s)v_{s,s}^{\theta}(I_s)\right|^2\right] ds dt, \qquad (26)$$

    *iii. The progressive self-distillation loss,*

$$\mathcal{L}_{\mathsf{PSD}}(\hat{v}) = \mathcal{L}_b(v^{\theta})$$
$$+ \int_0^1 \int_0^t \int_0^1 \mathbb{E}\left[\left|v_{s,t}^{\theta}(I_s) - \gamma v_{s,u}^{\theta}(I_s) - (1-\gamma)v_{u,t}^{\theta}(X_{s,u}^{\theta}(I_s))\right|^2\right] ds dt d\gamma, \qquad (27)$$

    *where $u = (1-\gamma)s + \gamma t$ with $\gamma \in [0,1]$.*

*Proof.* For completeness, and for ease of the reader, we sketch the proof from Boffi et al. (2025). We first observe that $\mathcal{L}_b(v^{\theta}) \geq \mathcal{L}_b(v)$ where $v_t(x) = \mathbb{E}[\partial_t I_t | I_t = x]$ is the ideal flow. From this, it follows that $\mathcal{L}_{\mathsf{LSD}}(\hat{v}) \geq \mathcal{L}_b(v)$, $\mathcal{L}_{\mathsf{ESD}}(\hat{v}) \geq \mathcal{L}_b(v)$, and $\mathcal{L}_{\mathsf{PSD}}(\hat{v}) \geq \mathcal{L}_b(v)$, because the second term in each loss is non-negative. Moreover, by Proposition 5, the ideal flow map satisfies the Lagrangian equation 14 and the Eulerian equation 15, so that the second term is zero for both $\mathcal{L}_{\mathsf{LSD}}(\hat{v})$ and $\mathcal{L}_{\mathsf{ESD}}(\hat{v})$ at the true flow map.

It remains to show that the second loss in equation 27 imposes the semigroup condition. To see this, we observe that we may write the semigroup condition as

$$X_{s,t}(x) = X_{u,t}(X_{s,u}(x)),$$
$$\iff x + (t-s)v_{s,t}(x) = X_{s,u}(x) + (t-u)v_{u,t}(X_{s,u}(x)),$$
$$\iff x + (t-s)v_{s,t}(x) = x + (u-s)v_{s,u}(x) + (t-u)v_{u,t}(X_{s,u}(x)),$$
$$\iff (t-s)v_{s,t}(x) = ((1-\gamma)s + \gamma t - s)v_{s,u}(x) \qquad (28)$$
$$+ (t - (1-\gamma)s - \gamma t)v_{u,t}(X_{s,u}(x)),$$
$$\iff (t-s)v_{s,t}(x) = \gamma(t-s)v_{s,u}(x) + (1-\gamma)(t-s)v_{u,t}(X_{s,u}(x)),$$
$$\iff v_{s,t}(x) = \gamma v_{s,u}(x) + (1-\gamma)v_{u,t}(X_{s,u}(x)).$$

From the sequence of operations in equation 28, we see that the second term in equation 27 penalises the square residual on a rescaled semigroup condition. The minimiser therefore satisfies the semigroup condition, and by Proposition 5 is the ideal flow map. $\qquad \square$

**Stopgradients.** In practice, it is beneficial, when optimising over neural networks, to use the $\mathrm{stopgrad}(\cdot)$ operator to control the flow of information. This allows us to interpret the flow model $v_{t,t}^{\theta}$ as a "teacher", which is used to provide a training signal for the map on the off diagonal $s \neq t$. If the squared residuals presented in Proposition 6 are minimized directly, gradients can align the trained flow model with the untrained flow map, rather than vice-versa. To this end, we provide some concrete recommendations inspired by the setting when a frozen pre-trained teacher model is available.

    1. For LSD, we recommend:

$$\mathcal{L}_{\mathsf{LSD}}(\hat{v}) = \mathcal{L}_b(\theta) + \int_0^1 \int_0^t \mathbb{E}\left[\left|\partial_t X_{s,t}^{\theta}(I_s) - \mathrm{stopgrad}(v_{t,t}^{\theta}(X_{s,t}^{\theta}(I_s)))\right|^2\right] ds dt, \quad (29)$$

2. For ESD, we recommend:

$$\mathcal{L}_{\text{ESD}}(\hat{v}) = \mathcal{L}_b(\theta) + \int_0^1 \int_0^t \mathbb{E}\left[\left|\partial_s X^\theta_{s,t}(I_s) + \text{stopgrad}(\nabla X^\theta_{s,t}(I_s)v^\theta_{s,s}(I_s))\right|^2\right] \mathrm{d}s\mathrm{d}t,$$

(30)

3. For PSD, we recommend:

$$\mathcal{L}_{\text{PSD}}(\hat{v}) = \mathcal{L}_b(\theta)$$
$$+ \int_0^1 \int_0^t \int_0^1 \mathbb{E}\left[\left|v^\theta_{s,t}(I_s) - \text{stopgrad}(\gamma v^\theta_{s,u}(I_s) + (1-\gamma)v^\theta_{u,t}(X^\theta_{s,u}(I_s)))\right|^2\right] \mathrm{d}s\mathrm{d}t\mathrm{d}\gamma,$$

(31)

where $u = (1-\gamma)s + \gamma t$ with $\gamma \in [0,1]$.

The recommendations in equation 29 and equation 31 are precisely what would arise given a pre-trained teacher. The recommendation in equation 30 is similar, but also places a $\text{stopgrad}(\cdot)$ on the spatial Jacobian of the model, which often improves optimization stability significantly.

**Mean Flows and consistency training**. Given the choice of $\text{stopgrad}(\cdot)$ in equation 30, the resulting parameter gradient will be *linear* in $v^\theta_{s,s}$. We may then replace $v^\theta_{s,s}(I_s)$ by the Monte Carlo estimate $\partial_t I_s$ of the ideal flow $b_s(x) = \mathbb{E}[\partial_t I_s | I_s = x]$, using the tower property of the conditional expectation $\mathbb{E}[f(I_s)\partial_t I_s] = \mathbb{E}[f(I_s)\mathbb{E}[\partial_t I_s | I_s]] = \mathbb{E}[f(I_s)v_s(I_s)]$ for any function $f : \mathbb{R}^d \to \mathbb{R}^d$. Without this choice of $\text{stopgrad}$, this replacement is not possible due to the quadratic term $\mathbb{E}[|\nabla X^\theta_{s,t}(I_s)\partial_t I_s|^2] \neq \mathbb{E}[|\nabla X^\theta_{s,t}(I_s)v_s(I_s)|^2]$ because of the nonlinearity in $|\cdot|^2$. With this $\text{stopgrad}$, the quadratic term is a constant from the perspective of the gradient and hence this discrepancy can be ignored.

This trick is used by both consistency training (Song et al., 2023a; Lu & Song, 2025) and Mean Flows (Geng et al., 2025); in fact, these algorithms can be recovered by expanding

$$\partial_s X^\theta_{s,t}(x) = -v^\theta_{s,t}(x) + (t-s)\partial_s v^\theta_{s,t}(x),$$
$$\nabla X^\theta_{s,t}(x) = I + (t-s)\nabla v^\theta_{s,t}(x).$$

(32)

Plugging equation 32 into the Eulerian loss equation 30 yields

$$\mathcal{L}(\theta) = \mathcal{L}_b(\theta)$$
$$+ \int_0^1 \int_0^t \mathbb{E}\left[\left|-v^\theta_{s,t}(I_s) + (t-s)\partial_s v^\theta_{s,t}(I_s) + \text{stopgrad}\left(v^\theta_{s,s}(I_s) + (t-s)\nabla v^\theta_{s,t}(I_s)v^\theta_{s,s}(I_s)\right)\right|^2\right] \mathrm{d}s\mathrm{d}t.$$

Replacing $v^\theta_{s,s}(I_s)$ by the Monte Carlo estimate of the ideal flow as described above yields

$$\mathcal{L}(v^\theta) = \mathcal{L}_b(v^\theta)$$
$$+ \int_0^1 \int_0^t \mathbb{E}\left[\left|-v^\theta_{s,t}(I_s) + (t-s)\partial_s v^\theta_{s,t}(I_s) + \text{stopgrad}\left(\partial_s I_s + (t-s)\nabla v^\theta_{s,t}(I_s)\partial_s I_s\right)\right|^2\right] \mathrm{d}s\mathrm{d}t,$$

(33)

and then choosing to $\text{stopgrad}$ the $\partial_s v^\theta_{s,t}$ term as well as the spatial gradient yields the Mean Flow/Consistency Training objective,

$$\mathcal{L}(v^\theta) = \mathcal{L}_b(v^\theta)$$
$$+ \int_0^1 \int_0^t \mathbb{E}\left[\left|v^\theta_{s,t}(I_s) - \text{stopgrad}\left((t-s)\partial_s v^\theta_{s,t}(I_s) + \partial_s I_s + (t-s)\nabla v^\theta_{s,t}(I_s)\partial_s I_s\right)\right|^2\right] \mathrm{d}s\mathrm{d}t.$$

(34)

## B PROOFS

### B.1 GENERALISED FLOW MAP CHARACTERIZATIONS

**Proposition 1.** *Let $X_{s,t}$ be parametrised as $X_{s,t}(x) = \exp_x((t-s)v_{s,t}(x))$. Then $X_{s,t}$ is the unique GFM for equation 2 if and only if it satisfies any of the following conditions:*

*(i) Generalised Lagrangian Condition:*

$$\forall (s,t) \in [0,1]^2, x_s \in \mathcal{M}, \qquad \partial_t X_{s,t}(x_s) = v_t(X_{s,t}(x_s)), \tag{3}$$

*(ii) Generalised Eulerian Condition:*

$$\forall (s,t) \in [0,1]^2, x_s \in \mathcal{M} \qquad \partial_s X_{s,t}(x_s) + \mathrm{d}(X_{s,t})_{x_s}[v_s(x_s)] = 0, \tag{4}$$

*(iii) Generalised Semigroup Condition:*

$$\forall (s,t,u) \in [0,1]^3, x_s \in \mathcal{M}, \qquad X_{u,t}\left(X_{s,u}(x_s)\right) = X_{s,t}(x_s). \tag{5}$$

To show that the the propositions hold, let us first demonstrate the legality of the above claims, that the compared vectors indeed belong to the same tangent planes. For the semigroup condition, observe that $X_{s,t}(x) \in \mathcal{M}$ for any $s$, $t$ and $x$, and therefore is evidently coherent.

**Lemma 2.** *For any $x \in \mathcal{M}$, $s,t \in [0,1]^2$, $\partial_t X_{s,t}(x) \in \mathcal{T}_{X_{s,t}(x)}\mathcal{M}$.*

*Proof of Lemma 2.* Recall that, for any $x, s, t$, $X_{s,t}(x) = \exp_x((t-s)v_{s,t}(x))$, and, without loss of generality, suppose that $s \leq t$. (The case $t > s$ is perfectly symmetric.) For $s = 1$, $t = 1$ and the proof is trivial, as $\vec{0} \in \mathcal{T}_x \mathcal{M}$. Let $s < 1$, and let $\tilde{X}_u(x) := X_{s,(1-s)u+s}(x)$, which coincides with $X_{s,t}(x)$ for $(1-s)u + s = t \iff u = \frac{t-s}{1-s} =: \tilde{u} \in [0,1]$. Remarking that $((1-s)u+s) \in [0,1]$ for all $0 \leq u \leq 1$, we have that $\tilde{X}$ defines a geodesic between $\tilde{X}_0(x) = x$ and $\tilde{X}_1(x) = X_{s,1}(x)$. Now, let $u < 1$, as $u = 0 \implies s = t$, from where the proof is also trivial. Therefore, it follows that $\partial_u \tilde{X}_u(x) = \log_{X_{s,u}(x)}(X_{s,1}(x))/(1-u)$, and considering the previous expression for $u = \tilde{u}$ finalises the proof from the definition of the logarithmic map. $\square$

The above validates the Lagrangian condition. Let us also justify the Eulerian condition: by Lemma 2, $\partial_s X_{s,t}(x_s) \in \mathcal{T}_{X_{s,t}(x_s)}\mathcal{M}$, and let us now prove that $\mathrm{d}(X_{s,t})_x[v_s(x)]$ is on the same tangent space.

**Lemma 3.** *For any $x \in \mathcal{M}$, $(s,t) \in [0,1]^2$, $\mathrm{d}(X_{s,t})_x[v_s(x)] \in \mathcal{T}_{X_{s,t}(x)}\mathcal{M}$.*

*Proof of Lemma 3.* This is true by definition. To see this, let $\gamma : [0,1] \to \mathcal{M}, \tau \mapsto \gamma(\tau)$ be a curve, such that $\gamma(0) = x$ and $\partial_\tau \gamma = v_s$. We therefore have that $X_{s,t} \circ \gamma : [0,1] \to \mathcal{M}$, and therefore $\partial_\tau (X_{s,t} \circ \gamma)(0) \in \mathcal{T}_{X_{s,t}(\gamma(0))}\mathcal{M} = \mathcal{T}_{X_{s,t}(x)}\mathcal{M}$. Knowing that $(X_{s,t})_x v_s(x) = \partial_\tau (X_{s,t} \circ \gamma)(0)$ concludes the proof. $\square$

We can now prove the main proposition.

*Proof of Proposition 5.* Let us first prove the Lagrangian condition. It comes naturally as

$$X_{s,t}(x_s) = x_t \implies \partial_t X_{s,t}(x_s) = \partial_t x_t = v_t(x), \tag{35}$$

which is true by definition, since $(x_t)_{t \in [0,1]}$ is a solution of Equation (2). As for the Eulerian condition, we note too that, for any $x \in \mathcal{M}$, $X_{s,t}(X_{t,s}(x)) = x$. (So, it is also invertible.) Taking the derivative through $s$,

$$\partial_s X_{s,t}(X_{t,s}(x)) = \partial_s x \tag{36}$$
$$\implies \partial_s X_{s,t}(X_{t,s}(x)) + (\mathrm{d}X_{s,t})_{X_{t,s}(x)}[\partial_s X_{t,s}(x)] = 0, \tag{37}$$

where the last line is due to the chain rule on manifolds. Moreover, by the probability flow ODE, we have that $\partial_s X_{t,s}(x) = v_s(X_{t,s}(x))$, and, letting $y = X_{t,s}(x)$, we have that

$$\partial_s X_{s,t}(y) + (\mathrm{d}X_{s,t})_y[v_s(y)] = 0. \tag{38}$$

Since $X_{s,t}$ is invertible for any $s$ and $t$ and defined on $\mathcal{M}$, it is bijective, and thus we can freely rename $y$ into $x$ to conclude that the Eulerian condition holds. Finally, the semigroup condition is trivially true from the definition: $(X_{u,t} \circ X_{s,u})(x_s) = x_t = X_{s,t}(x_s)$. $\square$

### B.2 PROOF FOR LEMMA 1

**Lemma 1.** *(Generalised Tangent Condition) Let $v_t(x_t) = \mathbb{E}_{x_t \sim \rho_t(x_t)}[\partial_t I_t \mid I_t = x_t]$ for any $t$ and $x_t \in \mathcal{M}$ be the drift of equation 2. Then it holds that $\lim_{s \to t} \partial_t X_{s,t}(x_s) = v_t(x_t)$.*

*Proof.* Proof of Lemma 1. This property follows from the Lagrangian condition. We have that $\partial_t X_{s,t}(x_s) = v_t(X_{s,t}(x_s))$. Taking the limit on both sides, we find that

$$\lim_{s \to t} \partial_t X_{s,t}(x_s) = \lim_{s \to t} v_t(X_{s,t}(x_s)) \tag{39}$$

$$\implies \lim_{s \to t} \partial_t X_{s,t}(x_s) = v_t \left( \lim_{s \to t} X_{s,t}(x_s) \right) = v_t(x_t), \tag{40}$$

where the last line comes from the continuity of the limit. $\square$

**Lemma 4.** *For any $(s,t) \in [0,1]^2$ and $x \in \mathcal{M}$, $\lim_{s \to t} \partial_t X_{s,t}(x) = v_{t,t}(x)$.*

*Proof.* Following elementary Riemannian geometry, we find that

$$\partial_t X_{s,t}(x) = \partial_t \exp_x((t-s)v_{s,t}(x)) = \mathrm{d}(\exp_x)_{(t-s)v_{s,t}(x)}(v_{s,t}(x) + (t-s)\partial_t v_{s,t}(x)). \tag{41}$$

Knowing that $\mathrm{d}(\exp_x)_{\vec{0}} = \mathrm{Id}_{\mathcal{T}_x \mathcal{M}}$, and taking the limit, we find the desired conclusion. $\square$

### B.3 GENERALISED SELF-DISTILLATION LOSSES

**Proposition 2.** *(Generalised Lagrangian self-distillation) The GFM is the unique minimiser of the objective in equation 7 for $\mathcal{L}_{\text{GFM-SD}}(\theta) = \mathcal{L}_{\text{G-LSD}}(\theta)$, where*

$$\mathcal{L}_{\text{G-LSD}}(\theta) = \mathbb{E}_{t,s,(x_0,x_1)} \left[ \left\| \partial_t X_{s,t}^{\theta}(I_s) - v_{t,t}^{\theta}(X_{s,t}^{\theta}(I_s)) \right\|_g^2 \right]. \tag{8}$$

*Proof.* The loss is zero if and only if $\partial_t X_{s,t}^{\theta}(I_s) = v_{t,t}^{\theta}(X_{s,t}^{\theta}(I_s))$ almost everywhere. We can conclude by Proposition 1. $\square$

**Proposition 3.** *(Generalised Eulerian self-distillation) The GFM is the unique minimiser of the objective in equation 7 for $\mathcal{L}_{\text{GFM-SD}}(\theta) = \mathcal{L}_{\text{G-ESD}}(\theta)$, where*

$$\mathcal{L}_{\text{G-ESD}}(\theta) = \mathbb{E}_{t,s,(x_0,x_1)} \left[ \left\| \partial_s X_{s,t}^{\theta}(x_s) + \mathrm{d}(X_{s,t}^{\theta})_{I_s}[v_{s,s}^{\theta}(I_s)] \right\|_g^2 \right]. \tag{9}$$

*Proof.* The loss is zero if and only if $X_{s,t}^{\theta}(x_s) - \mathrm{d}(X_{s,t}^{\theta})_{x_s}[v_s^{\theta}(x_s)] = 0$ almost everywhere. We can conclude by Proposition 1. $\square$

**Proposition 4.** *(Generalised Progressive Self-Distillation) The GFM is the unique minimiser over $v_\theta$ of equation 7 for $\mathcal{L}_{\text{GFM-SD}}(\theta) = \mathcal{L}_{\text{G-PSD}}(\theta)$ and $u \mid (s,t) \sim \mathcal{U}(s,t)$, where*

$$\mathcal{L}_{\text{G-PSD}}(\theta) = \mathbb{E}_{t,s,u,(x_0,x_1)} \left[ d_g^2 \left( X_{s,t}^{\theta}(I_s), X_{u,t}^{\theta} \left( X_{s,u}^{\theta}(I_s) \right) \right) \right]. \tag{10}$$

*Proof.* The loss is zero if and only if $X_{u,t}^{\theta}(X_{s,u}^{\theta}(x)) = X_{s,t}^{\theta}(x)$ almost everywhere. We can conclude by Proposition 1. $\square$

### B.4 CONNECTIONS TO EXISTING METHODS

#### B.4.1 GENERALISED MEAN FLOWS

Generalising Mean Flows directly is non-trivial as it requires defining the integral of a vector field on a manifold properly, which would involve parallel transport and therefore derivatives thereof. Also, Mean Flows operate on the vector field level as opposed to Flow Map Matching which operates on the level of the flow map. It is difficult to go from one level to another directly, as it will involve non-trivial curvature terms. Instead, we propose to heuristically follow our derivations in Appendix A, in the "stopgradients" section. Indeed, we can see that our loss involves the instantaneous vector field

of the modelled flow map, $v_{t,t}^\theta$, as opposed to the ideal flow $\partial_t I_t$; hence the use of the Levi-Civita connection along $v_s(I_s)$ instead of the differential evaluated at $v_{s,s}^\theta$:

$$\hat{\mathcal{L}}_{\text{G-MF}}(\theta) = \mathbb{E}_{t,s,(x_0,x_1)} \left[ \left\| v_{s,t}^\theta(I_s) - \text{stopgrad} \left( \partial_s I_s - (t-s) \nabla_{v_s(I_s)} v_{s,t}^\theta(I_s) \right) \right\|_g^2 \right], \quad (42)$$

which indeed recovers the Euclidean case as a special case. The Levi-Civita connection, $\nabla : \mathfrak{X}(\mathcal{M}) \times \mathfrak{X}(\mathcal{M}) \to \mathfrak{X}(\mathcal{M})$, $(v, X) \mapsto \nabla_v X$, is the unique torsion-free, bilinear, metric compatible connection on $(\mathcal{M}, g)$ that respects the Leibniz rule in $X$, and it defines a notion of covariant derivative for the vector fields on $(\mathcal{M}, g)$.

### B.4.2 GENERALISED SHORTCUT MODELS

Shortcut models (Frans et al., 2025) are exactly trained to enforce the semigroup property but on a discrete time grid. Expressed in our notation, the loss amounts to

$$\mathcal{L}(\theta) = \mathcal{L}_{\text{FM}}(\theta) + \mathbb{E}_{s,d,(x_0,x_1)} \left\| v_{s,s+2d}(x_s) - \text{stopgrad} \left( v_{s+d,s+2d}(x_s + dv_{s,s+d}(x_s)) \right) \right\|_2^2. \quad (43)$$

Indeed, $d$ can be seen as the time difference $t-s$, and, letting $d$ be uniformly distributed, we recover exactly Euclidean Progressive self-distillation, as noted in Boffi et al. (2024). The Riemannian case is strictly analogous, where the + operation is replaced by the exponential map, and the appropriate manifold distance is used.

## C   ADDITIONAL EXPERIMENTAL DETAILS

### C.1   THE RELEVANCE OF NLL

The NLL is a standard generative modelling benchmark for generative models (Chen & Lipman, 2024; Lipman et al., 2023; Song et al., 2020b). It is true, however, that in our context this metric does not evaluate directly the quality of our learnt GFMs. It is still relevant in at least two important ways:

1. It shows that the instantaneous vector field is still well-learnt, despite the self-distillation loss, which could have impacted the training dynamics on this part of the loss. We show that it does not. So, overall, our objective did not "sacrifice" the instantaneous vector field, and the sum of the self-distillation and flow matching losses are not inherently "incompatible".

2. The flow map and its learnt instantaneous vector field are approximations of one another. Therefore, while the likelihood of one is not *exactly* equal to that of the other, the difference between the two being typically very low (as measured by the self-distillation losses, down to $10^{-5}$ in MSE), the instantaneous vector field allows us to compute a good approximation of the likelihood induced by the flow map.

### C.2   EMPIRICAL MMD CALCULATION

For any distributions $p$ and $q$ with support $(\mathcal{M}, g)$, and with respective independent samples $(p_i)_{1 \le i \le n}$ and $(q_i)_{1 \le i \le n}$,

$$\widehat{\text{MMD}}(p,q) := \frac{1}{n^2} \sum_{i,j=1}^n \exp\left(-\kappa d_g^2(p_i, p_j)\right) + \exp\left(-\kappa d_g^2(q_i, q_j)\right) - 2\exp\left(-\kappa d_g^2(p_i, q_j)\right). \quad (44)$$

We choose $n$ to be equal to the size of the test set.

### C.3   ADDITIONAL RESULTS

Figure 7 includes additional (all) qualitative results on the Earth datasets.

### C.4   THE COMPUTATIONAL COST OF SELF-DISTILLATION

We include, here, Table 5 exhibiting the differences in training run-times (wall clock time) between our GFM objectives and plain RFM; the numbers reported have been gathered on the Volcano

Figure 7: Plots of densities for the various datasets and all compared methods. Depicted in red are the test-set samples. Datasets from left to right: volcano, earthquake, flood, fire.

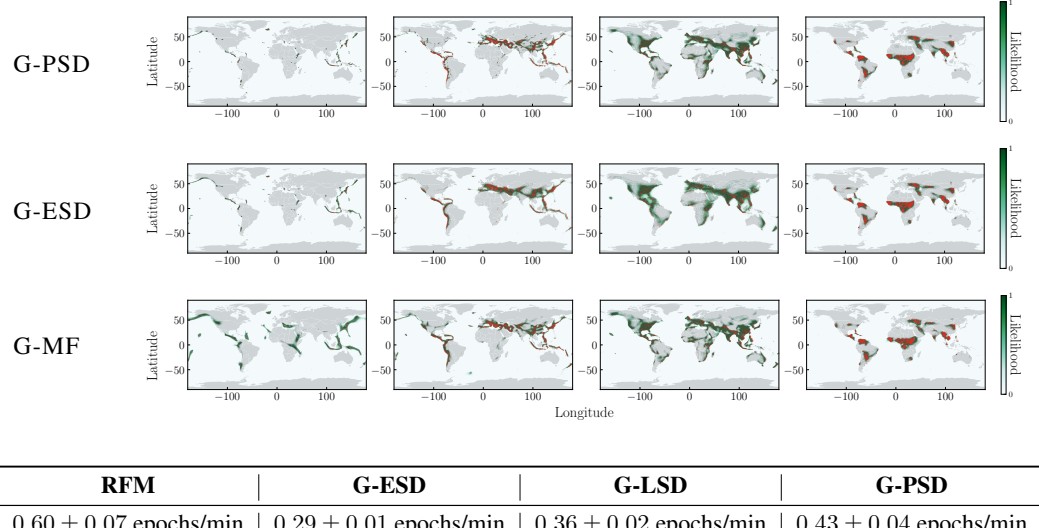

| RFM | G-ESD | G-LSD | G-PSD |
|---|---|---|---|
| $0.60 \pm 0.07$ epochs/min | $0.29 \pm 0.01$ epochs/min | $0.36 \pm 0.02$ epochs/min | $0.43 \pm 0.04$ epochs/min |

Table 5: Wall clock throughput time of our proposed methods on the volcano dataset (higher is faster).

dataset, which ran the largest model (and therefore is arguably most relevant; about 45M parameters). The gradation is rather intuitive: RFM has a simple flow matching loss; G-ESD computes a spatial Jacobian of a large mode; G-LSD only computes a time derivative of the same model; and G-PSD requires two additional forward passes without any gradients, compared to RFM.

