# OpenReview forum: "Generalised Flow Maps for Few-Step Generative Modelling on Riemannian Manifolds"
_ICLR.cc/2026/Conference — ICLR 2026 Poster_

### Official Review · Reviewer_73fW · 2025-10-17

**Soundness:** 3
**Presentation:** 3
**Contribution:** 3
**Rating:** 6
**Confidence:** 3

**Summary:**

This paper introduces Generalised Flow Maps (GFM), a unified framework for few-step generative modeling on Riemannian manifolds. It generalizes the notion of flow maps to non-Euclidean geometries such as spheres, tori, and Lie groups.
The core idea is to replace the Euclidean linear interpolant with geodesic-based Riemannian interpolation and to define manifold-consistent flow maps that jump between points along the probability flow ODE.
The authors derive three generalized self-distillation objectives, Generalised Lagrangian, Eulerian, and Progressive Flow Maps.
Empirically, GFM achieves state-of-the-art few-step generation on diverse geometric domains, such as protein torsion angles, geospatial data, and synthetic rotations.

**Strengths:**

* The paper extends flow-map-based generative modeling from Euclidean to general Riemannian manifolds, encompassing multiple geometric structures (tori, spheres, Lie groups, hyperbolic spaces).

* The paper is theoretically rigor.

* The paper tries to bridges recent few-step generation paradigms (Consistency Models, Mean Flows, Shortcut Models) with the geometric setting via G-LSD, G-ESD, and G-PSD objectives, which is a appealing story.

**Weaknesses:**

A potential weakness of this work lies in its limited empirical validation on large-scale or high-dimensional settings, where the proposed few-step framework would be most impactful. The observed gains are primarily evident at very few inference steps (as shown in Figures 3 and 4) and on relatively small datasets, making it unclear how well the method scales to more complex manifolds or realistic applications.
Unlike Euclidean diffusion or flow matching models, where few-step sampling is crucial for accelerating large-scale tasks such as protein generation or real-time image editing, the computational benefits in Riemannian domains remain less compelling.
In particular, for manifolds like torsion or angle spaces, existing diffusion or flow models already achieve fast inference, so the advantage of adopting a few-step Riemannian formulation is not yet convincingly demonstrated.

**Questions:**

NA

---

> ### Author Response · Authors · 2025-11-21
> **Rebuttal**
>
> We thank the reviewer for their time and feedback on our manuscript. We are happy to hear that the reviewer views our paper as generalising flow-map based generative models to Riemannian manifolds and that it is “theoretically rigorous”. Finally, we appreciate the reviewer for acknowledging that our paper unifies various few-step generative modelling attempts in one framework. We now answer the main questions raised by the reviewer below.
>
> ## Empirical validation
>
> We acknowledge the reviewer's concern that a few of the high-impact use cases for Geometric Generative Models remain in high-dimensional settings, such as protein backbone generation. We would like to first highlight that the utility of few-step generative models already shows incredible promise as we achieve SOTA results on existing lower-dimensional benchmarks, such as RNA torsion angles, model.
>
> To definitively answer the reviewer's question regarding scaling, we are presently running an experiment on protein backbone generation using the group $SE(3)^N$ and hope to report results as soon as possible and well before the end of the rebuttal period.
>
> > Unlike Euclidean diffusion or flow matching models, ..., the computational benefits in Riemannian domains remain less compelling.
>
> The Riemannian setting presents new complications that are unique to each manifold considered and are not a feature of Euclidean space models. For instance, hyperbolic manifolds are negatively curved and generally have led to numerically unstable modeling due to the fact that the exponential and logarithmic map effectively blow up as they use hyperbolic sinh and cosh functions [1, 2]. These computational tradeoffs are exacerbated if one follows the ODE trajectory as errors compound for each NFE. This is precisely an area where GFMs overcome these numerical challenges due to the fact that fewer NFEs are needed, and a few-step generative model is an ideal candidate.
>
>
> ## Concluding remarks
>
> We appreciate the reviewer for their time and detailed feedback. We are happy to answer any further questions the reviewer might still hold, and we are excited to provide our new experimental results on the larger-scale protein backbone generations imminently.
>
> ## References
>
> [1] Sala, Frederic, et al. "Representation tradeoffs for hyperbolic embeddings." International conference on machine learning. PMLR, 2018.
>
> [2] Yu, Tao, and Christopher M. De Sa. "Representing hyperbolic space accurately using multi-component floats." Advances in Neural Information Processing Systems 34 (2021): 15570-15581.

---

> > ### Comment · Reviewer_73fW · 2025-11-28
> > **Response to Authors' rebuttal**
> >
> > Hi, thank you for your rebuttal. I will keep my positive score unchanged and I'm looking forward to the additional experimental results.

---

### Official Review · Reviewer_qmRH · 2025-10-29

**Soundness:** 2
**Presentation:** 2
**Contribution:** 2
**Rating:** 4
**Confidence:** 4

**Summary:**

The paper introduces the Generalized Flow Maps (GFM) framework, which extends the flow map matching framework to Riemannian manifolds. It further generalizes the self-distillation loss from previous works to the Riemannian setting and demonstrates that each variant can be efficiently learned by neural networks. Experiments on RNA backbones, protein side-chains, and various toy datasets validate the empirical performance of the proposed methods.

**Strengths:**

1. The paper is well-written, and the proposed method is easy to follow and understand.
2. It extends the Flow Map Matching framework [1] to general Riemannian manifolds.


[1]. Boffi, Nicholas Matthew, Michael Samuel Albergo, and Eric Vanden-Eijnden. "Flow map matching with stochastic interpolants: A mathematical framework for consistency models." Transactions on Machine Learning Research (2025).

**Weaknesses:**

1. **Limited experimental scope.** The experiments are restricted to relatively simple toy datasets. As shown in previous works, Riemannian generative models have broad applications in various domains, such as material generation [1], molecular conformer generation [2], and protein backbone generation [3]. As mentioned in the paper, the inference processes of these models are often inefficient. Therefore, additional experiments on real-world applications would strengthen the paper’s contributions and demonstrate that the proposed methods can indeed improve the inference efficiency of Riemannian diffusion models.

2. **Applicability to general manifolds.** The proposed method appears to be applicable only to well-known manifolds with closed-form geodesics. In addition, it relies on an embedding of the manifold into Euclidean space. Consequently, for more complex manifolds without such prior geometric knowledge, the method would be difficult to apply in practice.

3. The results appear less innovative, as the proposed method can be seen as a direct generalization of existing approaches in Euclidean space [4, 5]. However, this would be acceptable if the method proves to offer meaningful benefits in real-world applications (see Weakness 1).


[1]. Miller, Benjamin Kurt, et al. "Flowmm: Generating materials with riemannian flow matching." arXiv preprint arXiv:2406.04713 (2024).

[2]. Jing, Bowen, et al. "Torsional diffusion for molecular conformer generation." Advances in neural information processing systems 35 (2022): 24240-24253.

[3]. Yim, Jason, et al. "SE (3) diffusion model with application to protein backbone generation." arXiv preprint arXiv:2302.02277 (2023).

[4]. Boffi, Nicholas Matthew, Michael Samuel Albergo, and Eric Vanden-Eijnden. "Flow map matching with stochastic interpolants: A mathematical framework for consistency models." Transactions on Machine Learning Research (2025).

[5]. Geng, Zhengyang, et al. "Mean flows for one-step generative modeling." arXiv preprint arXiv:2505.13447 (2025).

**Questions:**

1. Does the proposed method require the geodesics of the Riemannian manifold to have a closed-form expression? In addition, does it rely on an embedding of the manifold into a Euclidean space? I would like to confirm whether my understanding is correct.

---

> ### Author Response · Authors · 2025-11-21
> **Rebuttal Part 1/2**
>
> We are grateful to the reviewer for their time, effort, and dedication to the review process. We thank the reviewer for stating that our paper is “well-written” and that GFM as a method is “easy to follow and understand”. Furthermore, we appreciate that the reviewer agrees with the central claim in the paper as a strength, in that our work extends the Flow Map framework to general Riemannian manifolds. We now turn to the key clarification points raised in the review.
>
>
> ## Experimental scope
>
> We acknowledge the reviewer's concern that the experiments in our paper are presently demonstrated on lower-dimensional manifolds of interest. As the reviewer correctly points out, Riemannian generative models have been extended to other high-impact application domains such as materials and protein backbone generation—all of which fall under the $SE(3)$ Lie group that is common in the physical sciences. We note that, in contrast, several prior (seminal) works focused on general Riemannian manifolds to highlight the generality of their framework. This includes both the seminal Riemannian Diffusion Models [1,2], and the Riemannian Flow Matching papers [3], and our work and the experiments we conduct are on the same set of manifolds considered in these works. We argue that the breadth of manifolds considered in this work showcases the generality of proposing a **new generative model**, rather than focusing on a specific geometry in $SE(3)$.
>
> Moreover, note that the main difficulty in higher impact applications does not necessarily arise from the higher dimensionality of the manifold considered, arguably: for protein backbone generation, the manifold considered is $SE(3)^N$, where each of the manifolds in the product is “just” SE(3) (itself, a semidirect product of $SO(3)$ and 3D translations, the first of which we have shown to be achievable with GFM, and the other part being “merely” Euclidean). The main difficulty of the experiment arises in the size of the input ($N$) and how poorly standard **architectures** (which are relatively small) scale.
>
> Nevertheless, we take the reviewer's point as an opportunity to strengthen our paper, and as such, we are currently running experiments on protein backbone generation, and we hope to share positive results for GFM on this application domain imminently—well before the end of the rebuttal period.
>
> Part 1/2

---

> > ### Author Response · Authors · 2025-11-21
> > **Rebuttal Part 2/2**
> >
> > ## Applicability to General Manifolds
> >
> > The reviewer raises several interesting technical points in the applicability of GFM to manifolds without closed-form geodesics, as well as the need for Euclidean embedding—i.e., the extrinsic view of Riemannian geometry. We answer each point in turn.
> >
> > We first highlight that the first technical point is a shared pain point across **all Riemannian generative models**, including Riemannian Flow Matching and Riemannian Diffusion Models. In cases where the closed-form geodesic is not available, we must either simulate Euler-Lagrange equations to construct a noisy sample $x_t$ and perform Riemannian flow matching (RFM) by computing $\frac{\partial x}{\partial t}$ through automatic differentiation. In this case, as GFM also leverages RFM as a component of its loss, we are also easily able to accommodate this, and in particular, the Lagrange loss (LSD) and Progressive loss (PSD) self-distillation losses are the most natural to port as they require evaluation of $ v_{t,t} (x_t)$ and $X_{u,t}(X_{s,u}(x_s))$. Recent work has also demonstrated that in lieu of simulating the geodesic equations, we can also compute a stochastic estimate of the velocity for Riemannian flow matching by minimizing the kinetic energy of the path, which converges to a geodesic as a solution [4]. GFMs are also able to do this, but we leave this investigation as a direction for future work for exploring data-driven manifolds.
> >
> > Regarding the utilization of embedding in a Euclidean space. We first highlight that, due to the famous Nash embedding theorem (also Whitney), we are **always able to embed a Riemannian manifold into a higher-dimensional Euclidean space**. Thus, it is always advantageous to use the Euclidean embedding if the projection operators are easy to compute. We note that Riemannian Diffusion Models [1] and Riemannian Flow Matching [3], also solely exploit this perspective. It is also possible to operate using intrinsic coordinates—i.e., by gluing charts together on the manifold—and also learn to integrate along the probability flow ODE. GFMs are also able to do this, but for fair comparison with RDM and RFM, we exploit the same parametrization in reporting our results. We hope that this helps clear the great point raised by the reviewer.
> >
> > ## Novelty
> > We respect the reviewer's position that the novelty of our work should be contextualized, given the existing literature on flow maps in Euclidean space. We would like to very politely push back against this assertion by first highlighting—to the best of our knowledge—prior to our work, there was no existing literature on few-step generative models for Riemannian manifolds, outside of **concurrent work** on Riemannian Consistency Models [5], which appeared on arXiv after the submission deadline for ICLR. As a result, we argue that demonstrating few-step generative models, both the theory and practice leading to SOTA results on well-established manifolds, is a novel insight. This positions our paper as one of the first papers to unify few-step generative modelling on Riemannian manifolds, and remains a core contribution. Moreover, in our paper, our theory fully generalises the existing flow map models to Riemannian manifolds, as we demonstrate in our theoretical results in Appendix B, which contains Propositions 1-4 and Lemmas 2-4. These results make use of geometric identities such as the covariant derivative and require careful handling of geometric objects.
> >
> > To the reviewer's point on larger-scale experiments, we are in the process of finalizing a protein backbone generation experiment on $SE(3)^N$, which should enable the reviewer to more enthusiastically endorse our paper.
> >
> > ## Concluding remarks
> >
> > We thank the reviewer for their time and constructive feedback in their review. We believe we have successfully answered all the great points raised by the reviewer in this rebuttal. We encourage them to continue asking further questions if they have any—we are more than happy to respond. Otherwise, we would be encouraged if the reviewer would reconsider their assessment of our paper with this rebuttal in context, and also the forthcoming protein backbone experiments.
> >
> > ## References
> >
> > [1] Huang, Chin-Wei, et al. "Riemannian diffusion models." Advances in Neural Information Processing Systems 35 (2022): 2750-2761.
> >
> > [2] De Bortoli, Valentin, et al. "Riemannian score-based generative modelling." Advances in neural information processing systems 35 (2022): 2406-2422.
> >
> > [3] Chen, Ricky TQ, and Yaron Lipman. "Flow matching on general geometries." arXiv preprint arXiv:2302.03660 (2023).
> >
> > [4] Kapusniak, Kacper, et al. "Metric flow matching for smooth interpolations on the data manifold." Advances in Neural Information Processing Systems 37 (2024): 135011-135042.
> >
> > [5] Cheng, C., Wang, Y., Chen, Y., Zhou, X., Zheng, N., & Liu, G. (2025). Riemannian Consistency Model. arXiv preprint arXiv:2510.00983.

---

### Official Review · Reviewer_bSeb · 2025-10-31

**Soundness:** 3
**Presentation:** 4
**Contribution:** 3
**Rating:** 4
**Confidence:** 4

**Summary:**

The paper proposes Generalised Flow Maps (GFM), a novel framework for few-step generative modeling on Riemannian manifolds. It addresses the high computational cost of geometric diffusion and flow-matching models by directly learning the flow map (integrator), enabling single or few-step high-quality sampling. The approach generalizes existing Euclidean few-step models and achieves state-of-the-art sample quality across various geometric datasets.

**Strengths:**

GFM is tested on TNA torsion, Earth dataset, SO3, and Hyperbolic spaces. The exponential results are well proposed.

**Weaknesses:**

The experiments are demonstrated on relative low-dim spaces, unlike the Euclidean algorithms, e.g., Meanflow, that can be applied to large datasets.

In the case with relatively large NFE, e.g., NFE=8, there is almost no improvement. (Fig. 4)

**Questions:**

Why in Tab. 3 volcano dataset the proposed algorithm worse than several existing algorithms? Is it because lack of data since the volcano has only <1000 data.

The Euclidean version that requires fewer NFE algorithms, in my view, highly depends on the flat property of the space. Will the curved space itself lead to some difficulty in GFM? Do you have some intuition?

---

> ### Author Response · Authors · 2025-11-21
> **Rebuttal Part 1/2**
>
> We thank the reviewer for their time and effort in evaluating our manuscript. We also appreciate that the reviewer found our experimental results to be “well-proposed”, with the reviewer highlighting all of our diverse experimental datasets. We now answer the key clarifications raised by the reviewer.
>
> ## Larger dimensional datasets
>
> We appreciate the reviewer’s comment that generative models like MeanFlow have been applied to larger datasets in the Euclidean setting. We would like to politely push back on the need for larger dimensional experiments to validate the thesis of our paper: few-step inference on Riemannian manifolds. As the reviewer outlines in the strengths section of our paper, our GFM results are robustly tested on RNA Torsion angles, Earth datasets, SO(3), and hyperbolic manifolds, and achieve SOTA few-step generation results in comparison to the principal baseline, which is Riemannian Flow Matching. Nevertheless, we believe that including a larger-scale protein backbone generation experiment would strengthen and more strongly highlight our proposed GFM. To this end, we are currently training a GFM model on $SE(3)^N$ to generate protein backbones, and we will share these results imminently.
>
> ## Results with relatively large NFE
>
> We acknowledge the reviewer's point that when performing inference with large NFE, there is little improvement. We believe this is completely in line with intuition and the goals of this paper. GFM is designed to accelerate inference by performing “large” jumps along the probability flow ODE on a Riemannian manifold. If the GFM model is learned perfectly, one would be able to jump directly to the end point, and no amount of additional NFEs would improve this generation. Moreover, we argue that diminishing returns at NFE=8 already suggest that GFM learning is quite effective, as in many cases RFM requires significantly more ODE integration steps to achieve lower sample quality—as we demonstrate in all of our empirical results. Consequently, we believe this ablation demonstrates the power and strengths of GFM over Riemannian Flow Matching.
>
> Finally, note that the quality of the generation need not improve monotonously with the number of steps increasing, as noted by the authors of [3], since, instead, an accumulation of approximations may lead to higher deviation from the underlying “theoretical” flow.
>
> Part 1/2

---

> > ### Author Response · Authors · 2025-11-21
> > **Rebuttal Part 2/2**
> >
> > ## Questions
> >
> > > Why in Tab. 3 volcano dataset the proposed algorithm worse than several existing algorithms? Is it because lack of data since the volcano has only <1000 data.
> >
> > We value the reviewer's question. As the reviewer correctly observes, the Volcano dataset is by far the smallest of the four Earth geospatial datasets—$\approx 6 \times$ less data than the second smallest flood dataset. This results in a much higher variance in performance for all models, as evidenced by the larger standard deviations—with RFM being the largest with $\pm 1.67$---in contrast to the other Earth datasets. This also means that the learning problem for the generalised flow-map is more prone to these batch effects for computing the NLL which requires full trajectory inference. We further highlight that oftentimes NLL may not always accurately calculate the sample quality of generation [1]. In particular, we find that in Fig 4, when we plot MMD as a function of NFE, all GFM’s robustly beat RFM in the few-step regime and remain equal at high NFE. Finally, Fig 7 in Appendix C.2 also visualizes the generations of our GFM objectives, which show that the generations match the test distribution. These 2 additional results complement the Table 3 results and provide a more holistic evaluation.
> >
> > > The Euclidean version that requires fewer NFE algorithms, in my view, highly depends on the flat property of the space. Will the curved space itself lead to some difficulty in GFM? Do you have some intuition?
> >
> > This is a great question. Intuitively, the Euclidean version benefits from the **regularities/curvature of the probability flow ODE**, and not the flatness of the space itself. For instance, performing Euler integration along a perfectly straight line requires fewer NFEs than a very curved path, where more NFEs are needed to reduce error accumulation as the velocity field is changing more rapidly. In fact, this learnability is tied to the Lipschitz constant of the velocity field and impacts the learnability of the flow itself, with some cases demonstrating the impossibility of learnability due to a well-known phenomenon termed “mass teleportation,” which requires the velocity field to blow up to solve the generative modelling problem [2]. We highlight that this phenomenon is independent of the space itself, i.e., Euclidean/Riemannian, and depends on the nature of the transport problem.
> >
> > For the Riemannian settings we consider, the curvature of the space does not necessarily mean more NFEs are needed as it is manifested primarily through manifold operations such as the exponential and logarithmic maps. However, the use of such maps does impact numerical stability and learning dynamics—which remains true for **all Riemannian generative models** and not just GFMs.
> >
> >
> > ## Concluding remarks
> >
> > We greatly appreciate the reviewer's time and effort in this rebuttal period. We hope that through our responses and forthcoming larger-scale protein backbone experiments, we have robustly answered all the great points raised by the reviewer. If the reviewer is satisfied with our responses, we would be encouraged if the reviewer would consider a fresher evaluation of our manuscript with rebuttal answers in context. We are also available to answer any further questions that the reviewer may have.
> >
> > ## References
> >
> > [1] Jiralerspong, Marco, et al. "Feature likelihood divergence: evaluating the generalization of generative models using samples." Advances in Neural Information Processing Systems 36 (2023): 33095-33119.
> >
> > [2] Máté, Bálint, and François Fleuret. "Learning interpolations between boltzmann densities." arXiv preprint arXiv:2301.07388 (2023).
> >
> > [3] Boffi, Nicholas M., Michael S. Albergo, and Eric Vanden-Eijnden. "How to build a consistency model: Learning flow maps via self-distillation." arXiv preprint arXiv:2505.18825 (2025).

---

### Official Review · Reviewer_f4VY · 2025-11-01

**Soundness:** 2
**Presentation:** 3
**Contribution:** 3
**Rating:** 6
**Confidence:** 3

**Summary:**

This paper introduces Generalised Flow Maps (GFM), a new class of generative models for Riemannian manifolds that enables fast, few-step inference. It generalizes Euclidean flow map principles by proposing three self-distillation training methods: Generalised Lagrangian, Eulerian, and Progressive Flow Maps (G-LSD, G-ESD, G-PSD). These methods are derived from three equivalent theoretical conditions that characterize a manifold-constrained flow map. Experiments on various geometric datasets (tori, spheres, SO(3), hyperbolic) show GFMs achieve state-of-the-art sample quality in single- and few-step evaluations, outperforming traditional geometric generative models that require many simulation steps.

**Strengths:**

**Originality & Significance**: The paper's core contribution is the novel and significant generalization of flow map learning from Euclidean spaces to arbitrary Riemannian manifolds. This directly addresses the critical bottleneck of slow inference in existing geometric generative models.

**Quality & Clarity**: The work is technically sound, providing a rigorous theoretical foundation for the new GFM variants. The empirical validation is comprehensive, testing across a diverse set of non-trivial manifolds and demonstrating clear, state-of-the-art performance in few-step sampling. The paper is exceptionally well-written and easy to follow.

**Weaknesses:**

**Unexplained Performance Gaps**: The three GFM variants are derived from theoretically equivalent conditions but show vast empirical performance differences (e.g., G-LSD is far superior to G-ESD in Table 2). The paper notes this but lacks an in-depth analysis of why, which is a key practical limitation.

**Missing Training Cost Analysis**: The paper focuses exclusively on inference speed. However, the GFM objectives add a self-distillation loss to the standard RFM loss, implying a more expensive training procedure. This training cost trade-off is not discussed or quantified.

**Validity of NLL Metric**: The NLL is computed using the "instantaneous velocity" (the implicit vector field $v_{t,t}$). Given that GFMs are trained to learn the global flow map $X_{s,t}$ for few-step sampling, how valid is the NLL of the instantaneous flow as a primary measure of model quality?

**Missing Protein Backbone Experiments**: The introduction prominently features protein backbone generation ($SE(3)^N$) as a key high-impact application. Why were experiments on this crucial manifold omitted in favor of simpler torsion angle datasets?

**Questions:**

See **Weaknesses**.

---

> ### Author Response · Authors · 2025-11-21
> **Rebuttal Part 1/2**
>
> We thank the reviewer for their time, feedback, and constructive criticism of our work. We are heartened to hear that the reviewer finds the core contribution to be “novel” and a “significant generalization of flow map learning from Euclidean spaces”, addressing the “critical bottleneck of slow inference” in geometric generative models. We also appreciate that the reviewer found our paper to be “technically sound”, while providing a “rigorous theoretical foundation for the new GFM variants”. Finally, we are thrilled to hear that the reviewer found our “well-written and easy to follow”, and that our empirical validation to be “comprehensive”, and that it demonstrated “state-of-the-art” performance in few-step sampling. We now address the main clarification points raised in the review.
>
> ## Unexplained Performance Gaps
>
> We appreciate the reviewer's comment that the three GFM variants have equivalent theoretical motivations but differ in performance empirically. We begin by first highlighting that our findings mirror those of new work on self-distillation of flow maps in Euclidean spaces [1], (_i.e._, the LSD loss is generally the best). While we do not provide any theory on this matter, we posit that, compared to ESD, LSD avoids the spatial Jacobian, which is known to be problematic for large neural networks. This observation is further compounded in the Riemannian setting, as the Jacobian of the Generalised Flow Map—due to the curvature of the manifold and numerically unstable manifold operations—can quickly become difficult/unstable to model. As for the PSD objective, it avoids the spatial Jacobian, but it resembles continuous-time consistency models, which have well-documented training instabilities and high variance, even in the Euclidean setting [2, 3]. Moreover, and most importantly, it biases the $X_{s,t}$ to the distribution of $X_{ut} \circ X_{su}$, which is an **“imperfectly learnt” distribution**. Thus, in practice, this can lead to compounding errors due to distribution shift—i.e., the model was trained on the interpolation distribution and not on its own distribution. In contrast, the LSD objective avoids the pitfalls of ESD and PSD: it relies on a simple partial derivative on $t$, which is trained to match the instantaneous vector field, which is simultaneously trained in a flow matching fashion—thus, avoiding the introduction of bias in the objective. Critically, it also avoids numerically tricky manifold operations.
>
> ## Training cost analysis
>
> We acknowledge the reviewer's point that, while the benefits of GFMs are accelerated inference, their training burden computationally remains uncharacterized. The additional cost depends on the GFM variant, which we characterize in a new ablation on the Volcano dataset where we compute the training throughput measured in wall clock time (higher is faster). These results are included in a new updated Appendix C.4. and reproduced below for convenience.
> Intuitively, we have that RFM has a simple flow matching loss; G-ESD computes a spatial Jacobian of a large mode; G-LSD only computes a time derivative of the same model; and G-PSD requires two additional forward passes without any gradients, compared to RFM.
>
> Table: Training throughput in wall-clock time on the Volcanos dataset (higher is faster).
> | **RFM**            | **G-ESD**         | **G-LSD**          | **G-PSD**          |
> |--------------------|-------------------|---------------------|---------------------|
> | 0.60 ± 0.07 epochs/min | 0.29 ± 0.01 epochs/min | 0.36 ± 0.02 epochs/min | 0.43 ± 0.04 epochs/min |
>
>
>
> ## Validity of the NLL metric
>
> We value the reviewers' concern that the validity of the NLL hinges on $v_{t,t}$. First of all, note that the NLL metric is **computed exactly the same way as Riemannian Flow Matching**, and remains more broadly a _standard metric_ for generative model evaluation. It is not a measure of the quality of the few-step generation of our flow maps directly, but it is relevant in two important ways: It shows that the instantaneous vector field is still well-learned, despite the self-distillation loss, which could have impacted the training dynamics on this part of the loss. We show that it does not. So, overall, our GFM objectives do not “sacrifice” the instantaneous vector field, and the sum of the self-distillation and flow matching losses are not inherently “incompatible”.
> The flow map and its learnt instantaneous vector field are approximations of one another. Therefore, while the likelihood of one is not exactly equal to that of the other, the difference between the two being typically very low (as measured by the self-distillation losses, down to $10^{-5}$ in MSE), the instantaneous vector field allows us to compute a good approximation of the likelihood induced by the flow map.
> We thank the reviewer for pointing out the lack of clarification around the NLL, and we have included it in our revised version (see Appendix C.1).
>
> Part 1/2

---

> > ### Author Response · Authors · 2025-11-21
> > **Rebuttal 2/2**
> >
> > ## Protein Backbone Experiments
> >
> > We thank the reviewer for their suggestion. We are currently running these experiments and hope to share our findings very soon—and well before the end of this rebuttal period. We will update the paper with these results as soon as they are available.
> >
> > ## Concluding remarks
> >
> > We thank the reviewer again for their time, effort, and engagement during this review process. We hope that our clarifications above and forthcoming protein backbone experiments are definitive enough to definitively answer all the great points raised by the reviewer. We are also more than happy to engage in any further discussion should the reviewer deem it necessary.
> >
> > ## References
> >
> > [1] Boffi, Nicholas M., Michael S. Albergo, and Eric Vanden-Eijnden. "How to build a consistency model: Learning flow maps via self-distillation." arXiv preprint arXiv:2505.18825 (2025).
> >
> > [2] Lu, Cheng, and Yang Song. "Simplifying, stabilizing and scaling continuous-time consistency models." arXiv preprint arXiv:2410.11081 (2024).
> >
> > [3] Song, Yang, and Prafulla Dhariwal. "Improved techniques for training consistency models." arXiv preprint arXiv:2310.14189 (2023).
> >
> > [4] Zhang, Huijie, et al. "AlphaFlow: Understanding and Improving MeanFlow Models." arXiv preprint arXiv:2510.20771 (2025).

---

### Meta-Review · Area_Chair_xEjD · 2026-01-04

**Summary:**

**Summary of contribution** \
The paper proposes a generalization of Flow Maps models to geometric data defined on Riemannian manifolds. This extension enables fast data generation, requiring only single- or few-step function evaluations, while maintaining high fidelity to the underlying geometric structure of the data.

**Concerns** \
All reviewers appreciated the clarity of the paper and the contribution of extending Flow Maps models to the Riemannian setting. However, they consistently noted that the experimental evaluation is limited in scope. Given that the proposed approach aims to improve the scalability of diffusion-based models for geometric data, a more extensive large-scale experimental validation was expected. The authors indicated during the rebuttal that they would provide an additional experiment on protein backbone generation, but this experiment was not made available by the end of the rebuttal period.

Aside from this limitation, no other major concerns were raised.

**Decision** \
The paper constitutes a valuable addition to the literature on generative models for geometric data. Given that the experimental evaluation follows standard practices in the field, the submission can be considered sufficiently complete for acceptance. The additional experiments promised during the rebuttal would further strengthen the empirical evidence supporting the proposed generative framework and should ideally be included in the camera-ready version.

**Reviewer Concerns:**

Reviewer qmRH highlighted an important limitation of the proposed approach, namely that it is applicable only to manifold-valued data with closed-form geodesics. As acknowledged by the authors, this limitation is shared by several existing works in the area and naturally points to promising directions for future research.

Reviewer f4VY requested additional clarification regarding the choice of the negative log-likelihood as an evaluation metric. The authors provided a convincing justification and revised the manuscript accordingly. The same reviewer also asked for an analysis of the additional computational cost required to achieve single- or few-step function evaluations at inference time. The authors addressed this concern by including a corresponding analysis in a newly added section of the Appendix.

**Reviewer Scores:**

The main concern raised by the reviewers was not addressed by the authors during the rebuttal. As a result, reviewers would have either maintained their original scores or adjusted them only slightly in a positive direction. Nevertheless, given that the experimental evaluation follows standard practices in the field, the absence of an additional experiment does not constitute sufficient grounds for rejection. Moreover, the contribution of the paper is clearly articulated and demonstrates strong potential for future development.

---

### Decision · Program_Chairs · 2026-01-26

Accept (Poster)